# HyperTree Proof Search for Neural Theorem Proving

**Guillaume Lample**[*†]  **Marie-Anne Lachaux**[*†]  **Thibaut Lavril**[*†]  **Xavier Martinet**[*†]

**Amaury Hayat**[§]  **Gabriel Ebner**[‡]  **Aurélien Rodriguez**[†]  **Timothée Lacroix**[*†]

## Abstract

We propose an online training procedure for a transformer-based automated theorem prover. Our approach leverages a new search algorithm, HyperTree Proof Search (HTPS), that learns from previous proof searches through online training, allowing it to generalize to domains far from the training distribution. We report detailed ablations of our pipeline's main components by studying performance on three environments of increasing complexity. In particular, we show that with HTPS alone, a model trained on annotated proofs manages to prove $65.4\%$ of a held-out set of Metamath theorems, significantly outperforming the previous state of the art of $56.5\%$ by GPT-f. Online training on these unproved theorems increases accuracy to $82.6\%$. With a similar computational budget, we improve the state of the art on the Lean-based miniF2F-curriculum dataset from $31\%$ to $42\%$ proving accuracy.

## 1 Introduction

Over the course of history, the complexity of mathematical proofs has increased dramatically. The nineteenth century saw the emergence of proofs so involved that they could only be verified by a handful of specialists. This limited peer review process inevitably led to invalid proofs, with mistakes sometimes remaining undiscovered for years (e.g. the erroneous proof of the Four Colour Conjecture [1]). Some mathematicians argue that the frontier of mathematics has reached such a level of complexity that the traditional review process is no longer sufficient, envisioning a future where articles are submitted with formal proofs so that the correctness can be delegated to a computer [2].

Unfortunately, very few mathematicians have adopted formal systems in their work, and as of today, only a fraction of existing mathematics has been formalized. Several obstacles have hindered the widespread adoption of formal systems. First, formalized mathematics are quite dissimilar from traditional mathematics, rather closer to source code written in a programming language, which makes formal systems difficult to use, especially for newcomers. Second, formalizing an existing proof still involves significant effort and expertise (the formalization of the Kepler conjecture took over 20 person years to complete [3]) and even seemingly simple statements sometimes remain frustratingly challenging to formalize.

To write a formal proof, mathematicians typically work with Interactive Theorem Provers (ITPs). The most popular ITPs provide high-level "tactics" that can be applied on an input theorem (e.g. the initial goal) to generate a set of subgoals, with the guarantee that proving all subgoals will result in a proof of the initial goal (reaching an empty set means the tactic solves the goal). An example of a proof in Lean [4], an interactive theorem prover, is given in Figure 1 and the corresponding proof hypertree[3] is illustrated in Figure 5 of the Appendix.

---

[*]Equal contribution. Corresponding authors: `{glample,malachaux,tlacroix}@fb.com`
[†]Meta AI Research    [‡]Vrije Universiteit Amsterdam    [§]CERMICS École des Ponts ParisTech
[3]A hypergraph is a graph where an edge leads to a set of nodes that is potentially empty in our set-up. A hypertree, in this work, is a hypergraph without cycles. Formal definitions can be found in Appendix C.1

36th Conference on Neural Information Processing Systems (NeurIPS 2022).

```
1    import data.nat.basic
2  ∨ example (m n k : ℕ) (h₀ : n ≤ m) : n + k ≤ m + k := begin
3  │    induction k,
4  ∨ │  {
5  │  │    exact h₀                          First subgoal : n + 0 ≤ m + 0
6  │  │  },
7  ∨ │  {
8  │  │    rw nat.succ_le_succ_iff,          Second subgoal :
9  │  │    exact k_ih                          n + k ≤ m + k ⇒ n + k + 1 ≤ m + k + 1
10 │  │  }
11     end
```

Figure 1: **A simple proof of the statement** $n \le m \Rightarrow n + k \le m + k$ **in Lean.** The *induction* tactic reduces the initial statement to two subgoals, that can be solved independently.

In this paper, we aim at creating a prover that can automatically solve input theorems by generating a sequence of suitable tactics without human interaction. The backward procedure naturally suggests a simple approach where a machine learning model trained to map goals to tactics interacts with an ITP to build the proof of an input goal in a backward fashion. The automated prover builds a hypergraph with the theorem to be proved as the root node, tactics as edges and subgoals as nodes. The prover recursively expands leaves by generating tactics with our model until we find a proof of the initial theorem. A proof is then a hypertree rooted in the initial theorem whose leaves are empty sets.

Unlike Chess or Go, particular challenges arise for tree-search in theorem proving. First, the action space, i.e. the amount of possible "moves" in a given state, is infinite (there is an unlimited number of tactics that can be applied to a given theorem). This requires sampling possible actions from a language model for which training data is scarce. Moreover, if all tactics sampled at a goal fail, we have no information on what region of the probability space to sample next. Second, in the context of theorem proving, we need to provide a proof of all subgoals created by a tactic, whereas AlphaZero[5] for two player games is allowed to focus on the most likely adversary moves.

This paper presents an in-depth study of our approach to overcome these difficulties and the resulting model, Evariste. In particular, we make the following contributions:

- A new MCTS-inspired search algorithm for finding proofs in unbalanced hypergraphs.
- A new environment (Equations) to easily prototype and understand the behavior of the models we train and our proof search.
- A detailed ablation study and analysis of the different components used in our approach on three different theorem proving environments. We study how data is selected for training the policy model after a successful or failed proof-search, what target should be used to train the critic model, and the impact of online training vs. expert iteration.
- State-of-the-art performance on all analyzed environments. In particular, our model manages to prove over 82.6% of proofs in a held-out set of theorems from `set.mm` in Metamath, as well as 58.6% on miniF2F-valid [6] in Lean.

## 2   Related work

Automated theorem proving has been a long-standing goal of artificial intelligence, with the earliest work dating from the 1950s [7, 8]. We focus here on recent work closest to ours and defer additional related work to Appendix B.

**Neural theorem provers.**    Recent work applying deep learning methods to theorem proving [9–11] are the closest to this work and obtained impressive results on difficult held-out sets for Metamath and Lean. The main differences between their approach and ours are the proof-search algorithm we propose, the training data we extract from proof-searches and our use of online training compared to their expert iterations. Another similar approach, Holophrasm [12], uses a different tree-search algorithm while others [13, 14] learn the search policy along with the tactic model. Unlike previous studies that focus on a single proving environment (e.g. Metamath, Lean, or HOL-Light), we extensively study the performance of our prover on three different formal languages, and found that some conclusions significantly vary based on the considered environment.

**MCTS and two player games.** AlphaZero [5] demonstrated good performances on two player games, replacing the Monte-Carlo evaluations of MCTS [15] with evaluations from a deep neural network and guiding the search with an additional deep policy. These ideas have been applied to first order logic proving in Kaliszyk et al. [16] with gradient boosted trees as policy and value models. Theorem proving can be thought of as computing game-theoretic value for positions in a min/max tree: to prove a goal, we need one move (max) that leads to subgoals that are all proven (min). This has led to other tree search algorithms such as Proof Number Search [17] or a more recent version, using a neural estimate of proof size: Wu et al. [18]. Noticing heterogeneity in the arities of min or max nodes, we propose a search method that goes down simultaneously in all children of min nodes, such that every simulation can potentially result in a full proof-tree.

## 3 Online training from proof searches

In this section, we introduce our Hypertree Proof Search (HTPS) algorithm and describe how it is used to generate training data for our model. We then detail our online training method.

### 3.1 Hypertree Proof Search

Given a main goal $g$ to automatically prove, HTPS is the algorithm that interacts with a policy model $P_\theta$ and a critic model $c_\theta$, and the theorem proving environment to find a proof hypertree for $g$. Proof search progressively grows a hypergraph starting from $g$, iteratively repeating the three steps illustrated in Figure 2: selection, expansion and back-propagation. The main difference with other search algorithms is our parallel descent in all subgoals of a tactic. A proof is found when there exists a hypertree from the root to leaves that are empty sets.

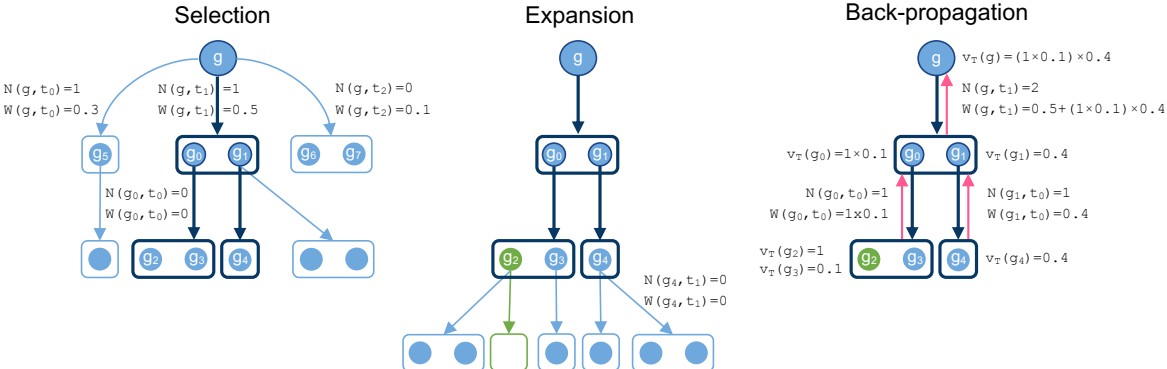

Figure 2: **HyperTree Proof Search**. We aim at finding a proof of the root theorem $g$. The figure represents the three steps of HTPS that are repeated until a proof is found. Proving either $\{g_5\}$, $\{g_0, g_1\}$, or $\{g_6, g_7\}$ would lead to a proof of $g$ by tactic $t_0$, $t_1$, or $t_2$. Guided by the search policy, we select a hypertree whose leaves are unexpanded nodes. Selected nodes are then expanded, adding new tactics and nodes to the hypergraph. Finally, we evaluate the node values $v_T$ of the hypertree starting from the leaves, using the critic, and back-propagating to the root, updating the visit counts $N$ and total action values $W$.

We assume a policy model $P_\theta$ and critic model $c_\theta$. Conditioned on a goal, the policy model allows the sampling of tactics, whereas the critic model estimates our ability to find a proof for this goal. Our proof search algorithm will be guided by these two models. Additionally, and similar to MCTS, we store the visit count $N(g, t)$ (the number of times the tactic $t$ has been selected at node $g$) and the total action value $W(g, t)$ for each tactic $t$ of a goal $g$. These statistics will be used in the selection phase and accumulated during the back-propagation phase of the proof search described below.

**Selection** The number of nodes in the proof hypergraph grows exponentially with the distance to the root. Thus, naive breadth-first search is infeasible to find deep proofs and some prioritization criteria is required to balance depth and breadth. Similar to *MCTS*, we balance the policy model's prior with current estimates from the critic. In particular, we experiment with two different search policies: PUCT [19] and Regularized Policy (RP) [20] (algorithms are detailed in Appendix C.3).

A key difference between previous work and ours is that our proof search operates on a hypergraph. Thus, whereas *MCTS* goes down a path from the root to an unexpanded node during its selection phase, our algorithm will instead create a partial proof hypertree, leading to a set of either solved or unexpanded nodes. To do so, we recursively follow the arg-max of the search policy from the root, until we reach the leaves of the current hypergraph (the detailed pseudo-code can be found in Section C.4). This selection step is illustrated in Figure 2.

In order to batch calls to the policy and critic models over more nodes to expand, we run several selections sequentially, using a virtual loss [21, 5] to produce different partial proof-trees. Note that solving all unexpanded leaves of any of these trees would immediately lead to a full proof of the root. In the next section, we describe how nodes are expanded.

**Expansion** To expand a node $g$, we use the policy model to sample tactics that would make progress on the goal. Tactics are sampled in an auto-regressive fashion (token by token) by the decoder [22], based on the previously generated tokens, and on a representation of the goal provided by the encoder. The generated tactics are then evaluated in the theorem proving environment. Each valid tactic will lead to a set of new subgoals to solve, or to an empty set if the tactic solves the goal. Finally, we add a hyperedge for each valid tactic $t_i$ from the expanded node $g$ to its (potentially empty) set of children for this tactic $\{g_i^0, ..., g_i^k\}$. Note that these children might already be part of the hypergraph. For new nodes, visit counts $N(g, t)$ and total action values $W(g, t)$ are initialized to zero. There are three types of nodes in the hypergraph:

- *Solved*: at least one tactic leads to an empty set, or has all its children solved.
- *Invalid*: all tactics sampled from the policy model were rejected by the environment, or lead to invalid nodes.
- *Unsolved*: neither solved nor invalid, some tactics have unexpanded descendants.

Note that the definitions for *Solved* or *Invalid* are recursive. These status are updated throughout the hypergraph anytime a hyperedge is added. Tactics leading to invalid nodes are removed to prevent simulations from reaching infeasible nodes. Once this is done, we back-propagate values from the expanded nodes up to the root, as described in the next section.

**Back-propagation** For each expanded goal $g$ in a simulated proof tree $T$, its value is set to $v_T(g) = 1$ if it is solved, and $v_T(g) = 0$ if it is invalid. Otherwise, its value is estimated by the critic model: $v_T(g) = c_\theta(g)$. This provides $v_T$ for all leaves of $T$ and we can then back-propagate in topographic order (children before parents) through all nodes of $T$. Interpreting the value of a node as the probability that it can be solved, since solving a goal requires solving all of its children subgoals, the value of a parent is the product of values of its children (we assume that the solvability of subgoals is independent, for simplicity):

$$v_T(g) = \prod_{c \in \text{children}(g,t)} v_T(c)$$

In particular, the value of a goal $g$ is the product of the values of all leaves in $T$ that remain to be solved to obtain a proof of $g$. Once all values in $T$ are computed, we increment the corresponding visit count $N(g, t)$ in the hypergraph as well as the total action values: $W(g, t) \mathrel{+}= v_T(g)$. For a goal $g$, the estimated value for tactic $t$ is then the mean of the total action value:

$$Q(g, t) = \frac{W(g, t)}{N(g, t)}$$

A fully detailed back-propagation step is illustrated in Figure 2.

### 3.2 Online training

Both the policy model $P_\theta$ and the critic model $c_\theta$ are encoder-decoder transformers [23] with shared weights $\theta$, which are trained online on two different objectives. The policy model $P_\theta$ takes as input a tokenized goal and generates tactics. It is trained with a standard seq2seq objective [22], where we minimize the cross-entropy loss of predicted tactic tokens conditioned on the input goal.

Our critic model $c_\theta$ is used to predict floating point values representing how likely a goal is to be solved. We start decoding with a special token, restrict the output vocabulary to two tokens PROVABLE and UNPROVABLE, and evaluate the critic with $c_\theta(g) = P(\text{PROVABLE}|g, \text{CRITIC})$. This objective is identical to a seq2seq objective where the cross-entropy is minimized over the two special tokens.

Our online training uses a distributed learning architecture reminiscent of AlphaZero [19] or distributed reinforcement learning setups [24, 5]. A distributed data parallel trainer receives training data from a set of asynchronous provers that run proof searches on theorems chosen by a controller. Provers, in turn, continuously retrieve the latest model versions produced by the trainers in order to improve the quality of their proof search. This set-up is represented in Figure 7 of the Appendix. Once a prover finishes a proof-search, we extract two types of training samples from its hypergraph:

**Tactic samples.** At the end of a successful proof search, we extract (goal, tactic) pairs of a minimal proof hypertree of the root node as training samples for the policy model. We use a different minimality criterion depending on the environment: number of proof steps for Metamath and Equations and total tactic CPU time for Lean. We show that this selection has a large impact on performances, other options such as selecting all solved nodes are investigated in Section 5.2.1. The policy model is trained with a standard seq2seq objective [22], where we minimize the cross-entropy loss of predicted tactic tokens conditioned on an input goal.

**Critic samples.** In the proof search hypergraph, we select all nodes that are either solved, invalid (all tactics failed or led to invalid nodes), or with a visit count higher than a threshold. Then, we use $c(g) = 1$ as the training target for solved nodes. For internal nodes, we use the final estimated action value $c(g) = W(g, t^*)/N(g, t^*)$ where $t^*$ is the tactic that maximizes the search policy at $g$. Finally, for invalid nodes, we use $c(g) = 0$.

The trainers receive training samples that are stored into two separate finite-size queues, one for each objective. When a queue is full, appending a new sample discards the oldest one. In order to create a batch for a task, we uniformly select samples in the corresponding queue.

During online training, in addition from this generated data, we also sample from the supervised datasets used for fine-tuning our models (see 4.1) which provide high-quality data. All training objectives are weighted equally. An overview of our full training pipeline is given in Appendix D.1.

Our proof-search depends on many hyper-parameters, and the optimal settings might not be the same for all statements, making tuning impractical. Thus, the controller samples these hyper-parameters from pre-defined ranges (see Appendix D.3 for details) for each different proof-search attempt.

# 4 Experiments

In this section, we provide details about our experimental training and evaluation protocols. We first describe the supervised datasets used to fine-tune our policy models, as well as the tokenization used. Then, we discuss the evaluation datasets and methodology. In Appendix D.2, we provide additional details about our model pretraining and architecture.

We develop and test our methods on three theorem proving environments: Metamath, Lean and Equations. Metamath [25] comes with a database of $35k$ human-written theorems called set.mm. We also evaluate our methods on the Lean proving environment, which provides a level of automation that is helpful to solve more complex theorems. Lean comes with a human-written library of $27k$ theorems called Mathlib [26].

## 4.1 Model fine-tuning and supervised datasets

Starting the HTPS procedure described in Section 3 from a randomly initialized model would be suboptimal, as no valid tactic would ever be sampled from the policy model. Thus, starting the online training from a non-trivial model is critical. To this end, we first fine-tune our policy model $P_\theta$ on a supervised dataset of theorems specific to each environment. We refer to this model as the *supervised* model.

**Metamath** In Metamath, we extract all proofs from the set.mm library, composed of 37091 theorems (c.f. Section D.5 for the version of set.mm). The training set is composed of around 1M

Table 1: **Dataset statistics for supervised training.**

|  | # train theorems | # train proof steps | Avg. goal length |
|---|---|---|---|
| Equations | $\infty$ | $\infty$ | 33.7 |
| Metamath | 35k | 1M | 120.1 |
| Lean | 24k | 144k | 169.3 |

goal-tactic pairs; more statistics about the training data are provided in Table 1. Tokenization in Metamath is trivial, as statements are composed of space-separated tokens.

**Lean**  Following [11], we extract a supervised dataset from the Mathlib library and co-train with the dataset of proof-artifacts of Han et al. [10] to reduce overfitting. To facilitate experimentation and reproducibility, we use fixed versions of Lean, Mathlib, and miniF2F (c.f. Appendix D.5). Finally, we add another supervised co-training task by converting to Lean a synthetic dataset of theorems generated by the Equations environment (c.f. Appendix E.6). Statistics about the training set are available in Table 1.

**Equations**  Finally, we developed a new environment, Equations, in the spirit of INT [27], as a simpler analogue to existing proving environments. Its expressivity is restricted to manipulating mathematical expressions (e.g. equalities or inequalities) with simple rules (e.g. $A + B = B + A$, or $A < B \Rightarrow -B < -A$). Unlike Metamath or Lean, the Equations environment does not come with a dataset of manually annotated proofs of theorems. Instead, we generate supervised data on the fly using the random graph generator described in Appendix E.5. As the model quickly reaches perfect accuracy on these synthetic theorems, we only leverage statements from the *Identities* split during online training.

## 4.2  Evaluation settings and protocol

In Polu et al. [11], the model is fine-tuned on theorems from the training set and expert iteration is done on theorems from different sources: train theorems, synthetic statements, and an extra curriculum of statements without proofs (miniF2F-curriculum). The produced model is then evaluated on unseen statements, namely the validation and test splits of the miniF2F dataset [6].

In this work, we also consider the *transductive* setup: on a corpus of unproved statements available at train time, how many proofs can our method learn to generate? This protocol is also sensible, as allowing the model to learn from a failed proof-search can lead to more focused exploration on the next attempt, proving more statements overall than a model that would not be trained online.

Following [9], we also evaluate the pass@k by running $k$ proof searches on the evaluated statements with the policy and critic obtained by online training. Given the many evaluations presented in this work, we only run them once. We give more details on the hyper-parameters used in Appendix D.3. In the transductive setup, we also report the *cumulative pass rate*, i.e. the proportion of theorems solved at least once during online training.

## 5  Results

In this section, we present our results and study the moving parts of our pipeline through ablations. We compare our results with GPT-f which represents the state of the art on Metamath and Lean.

Table 2: **Pass rate on Lean environment using 64 trials (pass@64)** Numbers with a $^\dagger$ exponent correspond to the cumulative pass-rate since the evaluated statements are part of the online training.

| Online training statements | Supervised - | GPT-f | Evariste-1d miniF2F-curriculum | Evariste-7d | Evariste miniF2F-valid |
|---|---|---|---|---|---|
| miniF2F-valid | 38.5 | 47.3 | 46.7 | 47.5 | $58.6^\dagger$ |
| miniF2F-test | 35.3 | 36.6 | 38.9 | 40.6 | 41.0 |
| miniF2F-curriculum | 20.8 | 30.6 | $33.6^\dagger$ | $42.5^\dagger$ | 32.1 |
| Train time (A100 days) | 50 | 2000 | 230 | 1620 | 1360 |

### 5.1 Main results

#### 5.1.1 Lean

In Lean, we run our experiments on A100 GPUs with 32 trainers and 200 provers. Each prover runs our Lean API on 48 CPU cores. Unlike Polu et al. [11], we sample statements equally from mathlib-train and miniF2F-curriculum, to avoid giving too much importance to statements from a different domain than the target. Results can be found in Table 2. After 1 day of training, each statement from miniF2F-curriculum has been sampled on average 250 times, and 110 out of the 327 statements have been solved. Our model outperforms GPT-f on miniF2F-test, with an approximately $10\times$ training time speed-up. After 7 days, we solve 139 statements of miniF2F-curriculum (100 for GPT-f), and observe further improvements on miniF2F-valid or miniF2F-test.

For other evaluations, we depart from the set-up of Polu et al. [11], directly using the statements from the miniF2F-valid split in our online training, obtaining 58.6% cumulative pass rate. We then evaluate the final model on miniF2F-test, reaching 41% pass@64, against 36.6% for GPT-f. Without the synthetic data co-training task, the performance drops to $54.9\%$ cumulative pass rate on the miniF2F-valid split, and $38.5\%$ pass@64 on the miniF2F-test split. Examples of proofs found by our model can be found in Appendix F.

#### 5.1.2 Metamath

On Metamath, we train our model on V100 GPUs, with 128 trainers and 256 provers, whereas ablations are run on 16 trainers and 32 provers. We report our results in Table 3 for the supervised model and for a model trained with online training. During online training, we sample statements from the training and from the validation splits of `set.mm` equally.

Online training dramatically improves performances on valid statements, going from a $61\%$ pass@8 to a cumulative pass rate of $82.6\%$. This improvement cannot solely be explained by the high number of attempts on validation theorems during training. Indeed, the ablation in Figure 3 (right) shows that Evariste significantly outperforms a supervised model with the same number of attempts. The supervised model plateaus at $66\%$ while Evariste keeps improving beyond $74\%$ after 7 days of training, showing that the model is able to learn from previous proof searches through online training.

On test theorems, for which statements were not provided during online training, the accuracy increased by $10\%$ compared to the supervised model, from $55.8\%$ to $65.6\%$ accuracy. The supervised model obtains a pass@32 accuracy of $65.4\%$ (resp. $61.2\%$) on the validation (resp. test) splits, compared to GPT-f's $56.5\%$ (resp. $56.2\%$) after expert iteration.

Table 3: **Results on Metamath for a supervised model and Evariste.** We report the pass@8 and pass@32 scores on the validation and test splits. Additionally, for Evariste we also report the cumulative score on the validation set, i.e. the fraction of theorems proved at least one time during online training. Note that for Evariste on Valid, the cumulative and pass@k performances are close since these statements were seen during training.

|  | Valid | | | Test | |
| --- | --- | --- | --- | --- | --- |
|  | cumulative | pass@8 | pass@32 | pass@8 | pass@32 |
| Supervised | N/A | 61.0% | 65.4% | 55.8% | 61.2% |
| Evariste | 82.6% | 81.0% | 81.2% | 65.6% | 72.4% |

#### 5.1.3 Equations

In Equations, we run our main experiment with 32 trainers and 64 provers, whereas ablations are run on 16 trainers and 32 provers. In this environment, the model easily learns the training distribution of our random generator, and solves all synthetically generated problems. Thus, online training is run on the *Identities* statements only. Our main experiment reaches a cumulative pass rate of $91.3\%$ on the *Identities* split, while a supervised model never exceeds $36\%$ even after a similar number of proof attempts. In Appendix 9, we give examples of *Identities* statements proved during online training, as well as the size and depth of proofs found by the model.

In particular, Evariste managed to find the proof of complex mathematical statements, such as $\sinh(x/2) = \sinh(x)/\sqrt{2(1+\cosh(x))}$ and $\tan(3x)(1 - 3(\tan(x))^2) = 3\tan(x) - (\tan(x))^2 \tan(x)$ that required 82 and 117 proof steps respectively, showing the abilities of HTPS to

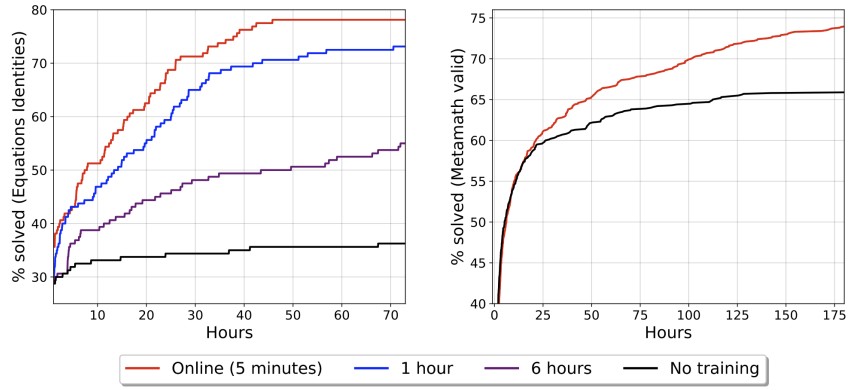

Figure 3: **Comparison between online setup, expert iteration, and fixed model.** We report the cumulative pass rate on the *Identities* (resp. valid) split on Equations (resp. Metamath). Reloading the model more frequently converges faster and to a better performance. When no training is done, the final performance is much lower despite using as many attempts, showing that online training is able to learn from previous proof searches.

prioritize subgoals and guide the search in very large proof graphs. This shows that online training is able to adapt our policy and critic models to a completely new domain, going from automatically generated statements to identities found in math books. Examples to understand the gap between these two domains can be found in Appendix E.

## 5.2 Ablation study

In this section, we present an ablation study on several components of our system. Since Lean experiments are CPU intensive, we run most of our ablations on the Equations and Metamath environments. On Lean, we ran experiments on a smaller subset of hyper-parameters that consistently performed well on the other environments.

### 5.2.1 Online training data for tactic objective

Table 4: **Performance of our model for different online training data for tactic objective.** We report the pass@8 score for Metamath and cumulative pass rate for Equations. We either keep all nodes and sample tactics according to the policy, or, extract (minimal) proofs of solved nodes, or (minimal) proofs of the root theorem only. Selecting minimal proofs always improves performance.

| Proof Of | All Solved | | Root | | All Nodes |
|---|---|---|---|---|---|
| Type of Proof | All | Min | All | Min | |
| Metamath (valid) | 61.2 | 65 | 57.4 | **68.6** | 51.6 |
| Metamath (test) | 57.2 | **58.8** | 54.8 | 57.4 | 54.4 |
| Equations (Identities) | 40.6 | **78.1** | 37.5 | 71.3 | 37.5 |

The way we filter tactics sent to the trainers has a large impact on final performances. We investigated several filtering methods and report the results in Table 4. The first method is similar to the one used in AlphaZero and exposed in [19]: we select all nodes of the proof search hypergraph where the visit count is above a certain threshold and we filter tactics above a given search policy score. At training time, tactics are sampled according to the filtered search policy. With this method the model reaches 51.6% pass@8 on the valid set of Metamath and 37.5% cumulative pass rate on Equations.

We then experimented with other filtering criteria, selecting only goal-tactic pairs that are part of proofs: either a proof of the root node, or of any solved node in the hypergraph. Then, we learn from all possible proofs, or only from proofs that are minimal according to a criteria (number of proof steps for Equations and Metamath, cumulative CPU time for Lean).

Learning only from minimal proofs always leads to improved performance, regardless of the selected roots. Learning from the minimal proofs of all solved nodes, we reach a cumulative pass rate of 78.1% on Equations, compared to 40.6% when learning from all proofs. On Metamath, only learning from the root's minimal proof gives the best result on the valid set, reaching a pass@8 of 68.6%.

### 5.2.2 Critic

Table 5: **Ablation study on the critic and search hyper-parameters in HTPS.** We report the pass@8 score for Metamath, and the cumulative pass rate for Equations. Evariste, trained with a soft critic and stochastic hyper-parameters, obtains the best performance in both environments. Removing the critic, or using a hard critic leads to reduced performances. In Equations, adding stochasticity in the proof search hyper-parameters increases the performance by $4.3\%$ in Equations, and slightly improves performance in Metamath.

|  | Evariste | No critic | Hard critic | Fixed search params |
|---|---|---|---|---|
| Metamath (valid) | 68.6 | 64.8 | 67.6 | **69.8** |
| Metamath (test) | **57.4** | 52.2 | 57.4 | 56.2 |
| Equations (*Identities*) | **78.1** | 65.6 | 63.1 | 73.8 |

To measure the impact of our critic model, we run an experiment where the proof search is only guided by the policy model. In particular, during the back-propagation phase, we set $v_T(g) = 0.5$ for leaves of $T$. In that context, our model is no longer trained with a critic objective. We report the results in Table 5. We find that using a critic model improves the performance significantly, by $5.2\%$ and $12.5\%$ on Metamath and Equations respectively.

As mentioned in Section 3, to train the critic objective, we set the training targets as $c(g) = 1$ for solved nodes, $c(g) = 0$ for invalid nodes and $c(g) = W(g, t^*)/N(g, t^*)$ where $t^*$ is the tactic that maximizes the search policy at $g$, for internal nodes. We also tested a hard critic estimation of the target values, following Polu and Sutskever [9], where $c(g) = 1$ for solved nodes and $c(g) = 0$ for both invalid and internal nodes. We report results in Table 5. For both Metamath and Equations, HTPS critic targets allow Evariste to reach its best performance. In Equations, the model reaches a cumulative pass rate of 78.1%, compared to 63.1% with hard critic estimates. In Equations, using hard critic targets gives worse performances than having no critic model at all, showing that these targets are a bad estimation: setting all internal nodes to zero is too pessimistic.

### 5.2.3 Fixed proof search parameters

We study the impact of sampling HTPS hyper-parameters for each attempt during online training. We run experiments with fixed, chosen search parameters for Equations and Metamath to compare with random sampling, and report results in Table 5. Evariste achieves better performances than the model trained with fixed search parameters on Metamath test set and Equations *Identities*, reaching 78.1% pass rate compared to 73.8% in Equations *Identities*.

### 5.2.4 Model update frequency during online training

In our online training procedure, the policy and critic models are updated every five minutes on the provers. We measure the impact of the frequency of these updates by trying different refresh rates: 5 minutes, 1 hour, 6 hours for Equations, and no updates at all for both Equations and Metamath. We report the cumulative pass rate over training hours in Figure 3. The higher the refresh rate, the better the cumulative pass rate over time, confirming the benefits of online training over expert iteration.

## 6  Conclusion

In this work, we introduce HTPS, an AlphaZero-inspired proof search algorithm for automated theorem proving, along with an online training procedure. We run an extensive study of our pipeline, and present state-of-the-art results on multiple proving environments. We show that online training provides large speed-ups over expert iteration, and allows generalization of the policy and critic models to completely new domains. Despite large number of attempts per theorem, proving the entirety of datasets like miniF2F remains elusive, and generating data from proof-search on the currently available corpora will likely be insufficient in the long term. As manually annotated formal datasets are limited, another way of providing exploration and additional training data (in the spirit of self-play for two player games) is required. Automated generation of new theorems is likely to be one of the future milestones.

## Acknowledgments

We thank the Meta AI and FLARE teams for useful comments and discussions throughout this work, notably, Baptiste Rozière, Faisal Azhar, Antoine Bordes, Quentin Carbonneaux, Maxime Darrin, Alexander Miller, Vincent Siles, Joe Spisak and Pierre-Yves Strub. We thank François Charton for his initial contributions to the Equations environment. We also thank Tristan Cazenave for interesting discussions around search algorithms, as well as the members of the Lean community for their help, notably Fabian Glöckle for valuable feedback on this project.

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
