# A  Proving environments

In this section we present the three proving environments used in this paper briefly. For each environment, we show a proof hypertree representation of a theorem, and give an example of tokenized goal and tactic from the training set.

## A.1  Metamath

Metamath's only rule is string substitution. Starting from a theorem to be proven, variables are substituted until we reach axioms. In our setup, we consider a tactic to be the label of a theorem in `set.mm`, along with the necessary substitutions. For instance, to show that $2 + 2 = 4$, we can use the rule `eqtr4i` which states that $A = B \land C = B \Rightarrow A = C$ with substitutions: $A = (2 + 2)$, $B = (2 + (1 + 1))$, and $C = 4$. We are then left with two subgoals to prove: $(2 + 2) = (2 + (1 + 1))$ and $4 = (2 + (1 + 1))$. The corresponding proof-tree can be found in Figure 4.

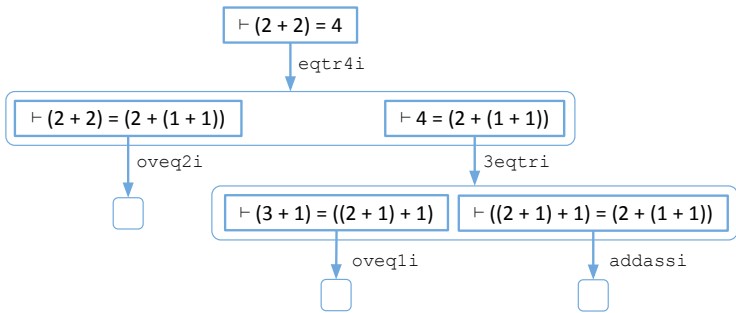

Figure 4: **A visualization of the proof-tree for $2 + 2 = 4$ in Metamath.**

The simplicity of Metamath makes it a great test bed for our algorithms. However, its lack of automation leads to larger proof sizes and its syntax and naming conventions make each step difficult to interpret for neophytes. Similar to GPT-f[9], we implement a parser for Metamath in order to automatically prove the syntactic correctness of statements. Moreover, we use this parser to allow generating only substitutions that cannot be inferred from the goal. The model is conditioned on a goal to prove, and is trained to output a sequence of the following format:



`LABEL MANDATORY_SUBSTS <EOU> PREDICTABLE_SUBSTS`



Below is a concrete example of tokenized goal along with its corresponding tactic. The applied rule is `ee10an`[4]. Since the values of `ph` and `th` can be directly inferred from the goal, we do not need generate them at training time, but we still use them during training to reduce overfitting.

```
<GOAL> |- ( A e. RR -> ( ( A - 2 ) + 2 ) = A ) </GOAL>

<TACTIC>
    ee10an
         ps = A e. CC 
         ch = 2 e. CC 
    <EOU>
         ph = A e. RR 
         th = ( ( A - 2 ) + 2 ) = A 
</TACTIC>
```

In order to speed-up decoding, we use a maximum decoding length of 512 tokens for Metamath which covers over $99\%$ of the human tactics in the supervised dataset.

---

[4]`https://us.metamath.org/mpeuni/ee10an.html`

## A.2 Lean

Lean is a full-fledged programming language and benefits from more powerful automation than Metamath, with tactics such as `ring` (able to prove goals using manipulations in semirings), `norm_num` (able to prove numerical goals) or `linarith` (able to find contradictions in a set of inequalities). Unlike Polu and Sutskever [9], our Lean API attempts to split tactic states into separate subgoals when no metavariable is shared. More details about our API and an example proof-tree can be found in Appendix D.4 and Figure 5.

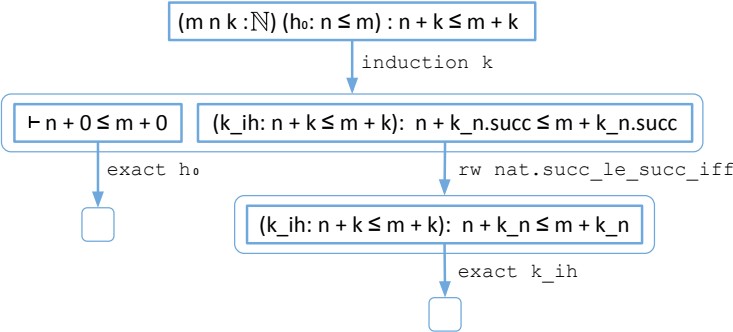

Figure 5: **A visualization of the proof-tree for the proof discussed in the introduction in Lean.**

Similar to Metamath, we use a maximum decoding length of 128 tokens which covers over 99% of the supervised human tactic dataset.

## A.3 Equations

We developed the Equations environment as a simpler analogue to existing proving environments. Its expressivity is restricted to manipulating mathematical expressions (e.g. equalities or inequalities) with simple rules (e.g. $A + B = B + A$, or $A < B \Rightarrow -B < -A$). This reduced expressivity makes goals and tactics easy to understand, helping with interpretability and debugging: plotting the set of goals explored during a Metamath proof search does not give a lot of insights on whether it is on track to find a proof. In Section E, we give an in-depth presentation of this environment.

Unlike in Metamath or Lean, we do not have access to a training set of human annotated proofs for this environment. Instead, we create a training set composed of randomly generated synthetic theorems and their proofs (see Section E.5 for details), and manually create an out-of-domain set of non-trivial mathematical identities for which we do not provide proofs. We refer to this evaluation split as *Identities*, a set of 160 mathematical expressions. As synthetic theorems randomly generated are much simpler and significantly differ from statements in the *Identities* split, we can evaluate the ability of our model to generalize to complex and out of domains data. An example proof-tree in Equations is shown in Figure 6.

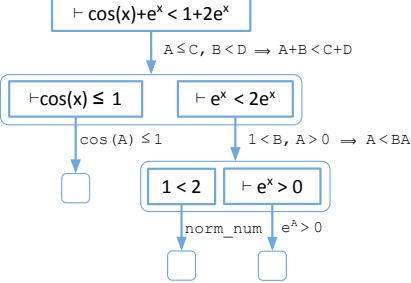

Figure 6: **A visualization of the proof-tree for** $\cos(\mathbf{x}) + \mathbf{e^x} < \mathbf{1} + \mathbf{2e^x}$ **in Equations.**

Below is an example tokenized goal with its tactic. The goal to prove is $((-(-((4 \times x4) + (x1 \times x4)))) <= (-(-((4 + x1) \times x4))))$, and is tokenized using the reverse Polish notation of the expression. The tactic factorizes the term in position 3, i.e. $(4 \times x4) + (x1 \times x4)$:

```
<GOAL> <= neg neg add mul + 4 x4 mul x1 x4 neg neg mul add + 4 x1 x4 <GOAL>
<TACTIC> ((A_*_C)_+_(B_*_C))|((A_+_B)_*_C) 3 <TACTIC>
```

Similar to Metamath and Lean, we limit the decoder size to 32 tokens to speed-up decoding.

## B Related works

Early approaches focused on simpler logics, culminating in extremely efficient first-order provers such as E [28] or Vampire [29]. However, these approaches are insufficient when it comes to theorems written in modern proof assistants such as Isabelle [30], Coq [31], or Lean [4]. Recently, the rising success of deep language models [32] and model-guided search methods [5] has spurred a renewed interest in the problem of automated theorem proving.

**Neural theorem provers** DeepHOL [33] focuses on the HOL-Light environment [34]. Their model relies on a classifier that can select among a restricted set of tactics and arguments, while we rely on a seq2seq model that can generate arbitrary tactics. The suggested tactics are then used in a breadth-first search. TacticToe [35] uses an MCTS without learned components, using ranking on predefined features to guide the search. Machine learning has also been used to improve classical provers by re-ranking clauses [36]. Finally, [37] uses a neural theorem generator to add data for policy and value training in Holophrasm [12].

**Reasoning abilities of language models.** Impressive performance of large language models in one or few shot learning [32], machine translation [22] or more recently code generation [38] spurred interest into the reasoning capabilities of large transformers. These model perform quite well on formal tasks such as expression simplification [39], solving differential equations [40], symbolic regression [41, 42], or predicting complex properties of mathematical objects [43]. These studies suggest that deep neural networks are well adapted to complex tasks, especially when coupled with a formal system for verification.

## C Hypertree Proof Search

### C.1 Hypergraph and definitions

We begin with some useful notations and concepts for our hypergraphs.

Formally, let $\mathcal{G}$ be a set of nodes, and $\mathcal{T}$ a set of tactics. A hypergraph is a tuple $H = (G, r, T, U)$ with $G \subset \mathcal{G}$ the nodes, $r \in G$ the root, and $T \subset G \times \mathcal{T} \times \mathcal{P}(G)$ the admissible tactics. An element of $T$ is written $(g, t, c)$ where $g$ is the start goal, $t$ is the applied tactic and $c$ is the potentially empty set of children that the tactic creates when applied to $g$ in the proving environment.

A hypertree is a hypergraph without cycles, i.e, such that we cannot find a path $g_0, \ldots, g_\ell = g_0$ with $\ell > 0$ and with $g_{i+1}$ in the children of $g_i$ for all $i$'s.

Let $S \subset G$ be the set of solved nodes. A node $g \in G \setminus U$ is solved if one of its tactic leads to no subgoals, or one of its tactics leads to only solved nodes. Formally: $\exists (g, t, \emptyset) \in T$ or $\exists (g, t, c) \in T$ such that $c \subset S$. We say that a tactic $t$ is *solving* for $g$ if all the children it leads to are solved. Conversely, let $U \subset G$ be the set of invalid nodes. A node $g \in G \setminus U$ is invalid if it has been expanded but has no tactics in the hypergraph, or all of its tactics have an invalid child. Formally: $\{(g, t, c) \in T\} = \emptyset$ or $\forall (g, t, c) \in T, c \cap I \neq \emptyset$.

These recursive definitions naturally lead to algorithms `MaintainSolved` and `MaintainStatus` to maintain sets $S$ and $I$ when elements are added to $H$.

A sub-hypertree $H_T$ of $H$ is a connected hypertree rooted at some goal of $H$. Its leaves $\mathrm{leaves}(H_T)$ are its subgoals without children (either elements of $U$ or $S$). The set of *proofs* of $g$ in $H$, $\mathrm{Proofs}(g, H)$ are all the hypertrees rooted at $g$ that have all their leaves in $S$. Similarly, the *expandable* subtrees of $H$ rooted in $g$, $\mathrm{Expandable}(g, H)$ are the subtrees with at least one leaf in $U$. A tactic is said to be *expandable* if it is part of an expandable subtree, this can be computed with a graph-search `ComputeExpandable`.

We can now reformulate the process of proof-search. Starting from a hypergraph that contains only the root theorem $r$, we produce a sequence of *expandable* subtrees. The unexpanded leaves of these subtrees are expanded in the hypergraph, then the new value estimates are backed-up. The hypergraph grows until we use all our expansion budget, or we find a proof of $r$.

## C.2 Algorithm

## C.3 Policies

When a goal $g$ is added to the hypergraph, its visit count $N(g,t)$ and total action value $W(g,t)$ are initialized to zero. Its virtual visit count $VC(g,t)$ are updated during proof search. Let $C(g,t) = N(g,t) + VC(g,t)$ be the total counts. These values are used to define the value estimate with a constant *first play urgency* [44]:

$$Q(g,t) = \begin{cases} \frac{\max(1,N(g,t))}{\max(1,C(g,t))} & \text{if } t \text{ is solving for } g \\ \frac{0.5}{\max(1,C(g,t))} & \text{if } N(g,t) = 0 \\ \frac{W(g,t)}{C(g,t)} & \text{otherwise.} \end{cases}$$

Notice that the value of solving tactics decreases with virtual counts, allowing exploration of already solved subtrees.

Given the visit count $N$, the total counts $C$, value estimates $Q$, the model prior $P_\theta$ and an exploration constant $c$. The policy used in Alpha-Zero is PUCT:

$$\text{PUCT}(g) = \arg\max_{t \in \mathcal{A}} \left[ Q(g,t) + c \cdot P_\theta(t|g) \cdot \frac{\sqrt{\sum N(g,\cdot)}}{1 + C(g,t)} \right]$$

Notice that more weight is given to the value estimate $Q$ as $N$ grows which decreases the second term. Another work [20] obtains good performances using as search policy the greedy policy regularized by the prior policy.

$$\pi_{RP}(g) = \arg\max_{y \in \mathcal{S}} \left[ Q(g)^T y - c \cdot \frac{\sqrt{\sum C(g,\cdot)}}{\sum(C(g,\cdot)+1)} KL(\pi_\theta, y) \right] \quad \text{with } \mathcal{S} \text{ the policy simplex at } g$$

Again, note that this policy balances the prior with the value estimates as the count grows, but does not account for individual disparities of visits of each tactics. In our experiments, we obtained better performances with $\pi_{RP}$ on Equations, and better performances with $PUCT$ on Metamath and Lean.

## C.4 Implementation details

**Simulation**  During simulation, we only consider subtrees that could become proofs once expanded. This means we cannot consider any invalid nodes or consider subgraphs containing cycles. If we encounter a tactic that creates a cycle during a simulation, this tactic is removed from the hypergraph, virtual counts from this simulation are removed and we restart the search from the root. This may remove some valid proofs, but does not require a backup through the entire partial subtree which would lead to underestimating the value of all ancestors. Removing tactics from the hypergraph also invalidates computations of *expandable* tactics. This is dealt with by periodically calling `MaintainExpandable` if no valid simulation can be found. A full description of the algorithm that finds one expandable subtree is available in Algorithm 1. Selection of nodes to expand requires finding expandable subtrees until a maximum number of simulations is reached, or no expandable tactic exists at the root. In addition to $W$, $N$ and $v_T$, we maintain a virtual loss counter $VC$ following Chaslot et al. [21], Silver et al. [5], so that successive simulations select different leaf subsets. This counter is initialized to zero for all nodes.

**Expansion**  The policy model produces tactics for an unexpanded node $g$. These tactics are evaluated in the proving environments. Valid tactics are filtered to keep a unique tactic (e.g. the fastest in Lean) among those leading to the same set of children. Finally, we add the filtered tactics and their children to the hypergraph. If no tactics are valid, the node is marked as invalid and we call `MaintainInvalid`. If a tactic solves $g$, the node is marked as solved and we call `MaintainSolved`.

---
**Algorithm 1** Finding an expandable subtree
---

**Input:** A hypergraph $H$ and its $root$
**Output:** A partial proof tree with unexpanded leaves
*:start*
T: hypertree($root$)
to_explore: list = [$root$]
**while** to_explore **do**
  g = to_explore.pop()
  **if** g is internal **then**
    **if** expandable($g$) $\neq \emptyset$ **then**
      tactic = arg $\max_t \pi\big|_{\text{expandable}(g)} \pi(g, t)$
    **else**
      continue { expandable nodes are in a sibling branch }
    **end if**
    **if** tactic leads to cycle **then**
      kill tactic
      remove virtual counts for elements of T
      goto *start*
    **end if**
    $VC(g, tactic)$ += 1
    T.add(g, tactic, children(g, tactic))
    to_explore += [children(g, tactic)]
  **end if**
**end while**

---

---
**Algorithm 2** Back-propagation of total action value $W$
---

**Input:** Partial proof-tree $T$ and value estimates $c_\theta(g)$ of its leaves.
to_backup = []
**for** $g$ in leaves of T **do**
  $v_T(g) = c_\theta(g)$
  to_backup.append((parent$_T(g)$, parent_tactic$_T(g)$))
**end for**
**while** to_backup **do**
  $g, t$ = to_backup.pop()
  to_update = $\prod_{c \in \text{children}_T(g)} v_T(c)$
  $W(g, t)$ += to_update
  $N(g, t)$ += 1
  $VC(g, t)$ -= 1
  $v_T(g)$ = to_update
  $g$.is_prop = true
  **if** all $c$.is_prop for $c$ in siblings$_T(g)$ **then**
    to_backup.append((parent$_T(g)$, parent_tactic$_T(g)$))
  **end if**
**end while**

---

**Backup** The backup follows topological order from the leaves of a simulated partial proof-tree $T$, updates $W$ and $N$, and removes the added virtual count. The algorithm is described in Algorithm 2

### C.5 Comparison with other search algorithms

**Best First Search** Several best-first search variations have been suggested for theorem proving. Proof number search (PNS) [17] is a best-first search algorithm that maintains the minimum number of expansion required to prove or disprove a node. Recent extensions have been proposed, including using a model for estimating the remaining proof / disproof number of a newly expanded node [18]. Similarly, Polu et al. [11] prioritize estimated proof-size in their best-first search objective.

Our node selection heuristic differs slightly: our critic gives the probability of solving a leaf, but our updates to $W(g)$ sums these log-probabilities and thus includes the proof-number information. Moreover, the arities at AND and OR nodes in our cases are highly unbalanced (up to 32 tactics at OR nodes, but very few children per tactic), selecting all children at AND nodes is computationally feasible (which is not the case in games where this would lead to an exponential growth of states to expand). Thus, we depart from the standard best-first search by expanding full candidate proof-trees at once.

**Monte Carlo Tree Search [15].** MCTS has been famously used as part of AlphaZero [19] to obtain great performances on two player games. This two player set-up can be mapped to theorem-proving by assigning one player to choosing the best tactics while the other player picks the most difficult goal to solve (a method explored in Holophrasm [12]). However, since we need to provide a proof of the root theorem, we need to ensure that we can solve *all* goals that a tactic leads to. This set-up has been studied for two player games when attempting to compute the game-theoretical value of positions. Using MCTS in this set-up is suboptimal [45], ignoring unlikely but critical moves from the opponent (in our case, a subgoal that looks easy but is impossible to solve). We decided to exploit the highly asymmetrical arities of our two players (most tactics lead to one or two goals) which makes simulating partial proof-trees computationally feasible. Thus, the values we back-propagate always take into account all possible moves from the opponent, while only requiring a few expansions per simulation.

**Polu and Sutskever [9]** This best-first search expands goals one at a time according to a priority-queue of either a value model or the cumulative log-prior from the language model. Since the priority is equal among siblings but strictly decreasing with depth, this means siblings will always be expanded together. However, nothing prevents the algorithm from jumping from one potential proof-tree to another, and potentially favoring breadth over depth. In comparison, depth does not appear in the value estimate we compute, but rather the remaining number of nodes to solve a particular proof-tree. Moreover, our algorithm leads to value estimates that can be used to train our critic, which performs better than 0-1 estimates provided by best-first search (c.f. Section 5.2.2).

## D Training details

### D.1 Full training pipeline

In order to bootstrap our online learning procedure we require a policy model $P_\theta$ that outputs coherent tactics. While the critic is left untrained, the policy model is fine-tuned on a pretrained transformer using a supervised dataset specific to the target environment. The full training pipeline can be summarized as follows:

- **Pretraining** of the encoder-decoder model on a large unsupervised corpus (c.f. Section D.2).
- **Fine-tuning** of the policy model on supervised datasets detailed in (c.f. Section 4.1).
- **Online training** of both the policy and critic models on data extracted from proof search (illustrated in Figure 7).

### D.2 Model architecture and training

**Model architecture.** Our transformer architecture uses a 12-layer encoder and a 6-layer decoder in all experiments. We use an embedding dimension of 1600 in the encoder and 1024 in the decoder

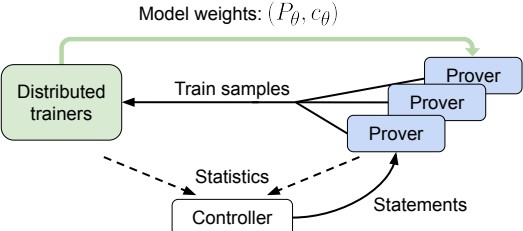

Figure 7: **An overview of our online training architecture.** The controller sends statements to asynchronous HTPS provers and gathers training and proving statistics. The provers send training samples to the distributed trainers and periodically synchronize their copy of the models.

for both Metamath and Lean. For Equations, where we expect the model to require less decoding capacity, the decoding dimension is lowered to 512. We found that reducing the decoder capacity increases the decoding speed without impacting the performance, as previously observed by Kasai et al. [46] in the context of machine translation. This observation led us to use "encoder-decoder" architecture and not a "decoder only" model (as in Polu and Sutskever [9]), in order to store most of the model capacity in the encoder and to sample tactics efficiently with a small decoder. Our models are composed of 440M parameters for Equations and 600M parameters for Metamath and Lean (for comparison, GPT-f uses a 770M parameter, 36-layer model).

**Model pretraining.** Model pretraining can be critical in low-resource scenarios where the amount of supervised data is limited [47, 48]. Thus, we do not immediately fine-tune our model but first pretrain it on a large dataset to reduce overfitting and improve generalization. In particular, we pretrain our model with a masked seq2seq objective (MASS [49]) on the LaTeX source code of papers from the mathematical section of arXiv. After tokenization, our filtered arXiv dataset contains around 6 billion tokens for 40GB of data. Similar to Polu and Sutskever [9], we observed large performance gains using pretraining. However, we found that arXiv alone provides a better pretraining than when it is combined with other sources of data (e.g. GitHub, Math StackExchange, or CommonCrawl).

**Supervised fine-tuning.** During fine-tuning, we train our models with the Adam optimizer [50] and an inverse square-root learning rate scheduler [23]. We use a dropout of 0.2 [51] to reduce the overfitting of our models. We also apply layer-dropout [52] with a dropout rate of 0.1 to further reduce overfitting and stabilize training. We implement our models in PyTorch [53] and use float16 operations to speed up training and to reduce the memory usage of our models.

**Online training.** During online training, we alternate between the *goal-tactic* objective, used during fine-tuning on the supervised dataset, and the *goal-tactic* and *goal-critic* objectives on data generated by the provers. As the model and the data generated by the provers are constantly evolving, we do not want the learning rate to decrease to 0, and we fix it to $3 \times 10^{-5}$ after the warm-up phase. Unless mentioned otherwise (e.g. for large experiments), we run all Metamath and Equations experiments with 16 trainers and 32 provers for a total of 48 V100 GPUs.

### D.3 Proof search hyper-parameters

HTPS depends on many hyper-parameters: the decoding hyper-parameters of the policy model and the search hyper-parameters. Selecting their optimal values would be difficult in practice, if not impractical, for several reasons. First, the model is constantly evolving over time, and the optimal parameters may evolve as well. For instance, if the model becomes too confident about its predictions, we may want to increase the decoding temperature to ensure a large diversity of tactics. Second, even for a fixed model, the ideal parameters may be goal-specific. If an input statement can only be proved with deep proofs, we should favor depth over breadth, and a small number of tactics per node. If the proof is expected to be shallow and to use rare tactics, we will want to penalize the exploration in depth and increase the number of tactics sampled per node. Finally, there are too many parameters to tune and running each experiment is expensive. Thus, we do not set HTPS hyper-parameters to a fixed value, but sample them from pre-defined ranges at the beginning of each proof. These pre-defined ranges were set *a priori* and were not tuned over the course of the experiments.

The decoding parameters and the chosen distribution are the following:

- **Number of samples:** the number of tactics sampled from the policy model when a node is expanded. Distribution: uniform on discrete values $[8, 16, 32, 48]$.
- **Temperature:** sampling temperature used during decoding. Distribution: uniform on range $[0.8, 2.0]$.
- **Length penalty:** penalty on the length of generated sequence. Distribution: uniform on range $[0, 1.2]$.

For the search parameters we have:

- **Number of expansions:** the search budget, i.e. the maximum number or nodes in the proof graph before we stop the search. Distribution: log-uniform with range $[1000, 10000]$.
- **Depth penalty:** an exponential value decay during the backup-phase, decaying with depth to favor breadth or depth. Distribution: uniform on discrete values $[0.8, 0.9, 0.95, 1]$.
- **Exploration:** the exploration constant $c$ in the policy (PUCT or RT). Distribution: log-uniform with range $[0.01, 100]$.

When sampling proof search parameters during evaluation, we use the same distributions than at training time, with two differences: we fix the number of expansions to 5k in Lean and 10k in Metamath.

## D.4 Details on our Lean API

States are more complex in Lean than in Metamath: metavariables can appear which are holes in the proof to be filled later. Subgoals sharing a metavariable cannot be solved in isolation. This is addressed in Polu and Sutskever [9] by using as input the entire tactic state. Instead, we inspect tactic states to detect dependencies between subgoals, and split the tactic state into different subgoals where possible in order to maximize state re-use and parallelization in the proof search algorithm. We only ever split the tactic state into contiguous lists of subgoals to make exporting the final proof easier.

Lean's kernel type checker has to be called after each tactic application as tactics sometimes generate incorrect proofs and rely on the kernel for correctness. For every goal in the previous tactic state, we type check the proof term inserted by the tactic. Since the kernel does not support metavariables, we replace every metavariable by a lambda abstraction.

## D.5 Metamath Lean versions

To compare our models in the same setup while working on this project, we ran all our experiments with a fixed version of Metamath and Lean. In particular, all experiments were run with the following GitHub commits of `set.mm`, Lean, miniF2F, and Mathlib:

- `https://github.com/metamath/set.mm`: 861bd3552636dcdb9cbc8df59d01b14520c72f82
- `https://github.com/leanprover/lean/`: tag/v3.3.0
- `https://github.com/openai/miniF2F`: 21723db70bbd030e034ed374db74cea4be1bf681
- `https://github.com/openai/miniF2F/tree/statement_curriculum_learning`: c9d827c871aff2ab0f5ec64a0d72e61111a7f072
- `https://github.com/leanprover-community/mathlib`: 9a8dcb9be408e7ae8af9f6832c08c021007f40ec

# E Equations environment

In this section, we give additional details about the environment Equations. First, we described its main elements, theorems (resp. tactics) in Section E.1 (resp. E.2). Then, we describe a proof in this environment in Section E.3, how numerical expressions are evaluated and what vulnerabilities this can lead to in Section E.4. Finally, we describe our random theorem generator in Section E.5 and how theorems and their proofs can be translated to Lean in Section E.6.

### E.1 Theorems

Each theorem in Equations consists in proving mathematical expressions composed of functions of real numbers, by manipulating and rewriting expressions. A theorem to prove can be an inequality or an equality, conditioned to a set of (potentially empty) initial assumptions. For instance:

$$x^2 + 1 \geq 2x \quad \text{or} \quad x > y \implies e^{y-x} - 1 < 0$$

In the first example, the goal does not have any hypothesis and consists in proving that for every $x \in \mathbb{R}$, $x^2 + 1 \geq 2x$. In the second example, the goal consists in proving that $e^{y-x} - 1 < 0$ for every $x, y \in \mathbb{R}$ that satisfy the hypothesis $x > y$.

Equalities and inequalities are represented as trees with the three following elements:

- **Leaves:** represent a variable, an integer, or a constant (e.g. $\pi$).
- **Internal nodes:** represent unary or binary operators, e.g. $+$, $-$, $/$, $\times$, $\exp$, $\ln$, $\cos$, $\sin$, $\sinh$, $\cosh$, etc. More advanced operators such as $\gcd$, $\mathrm{lcm}$, $\mathrm{mod}$ (the rest of an euclidean division) are possible when dealing with integers.
- **A root node:** represents a comparison operator, e.g. $=$, $\leq$, $<$, $\geq$, $>$, $\neq$. More advanced comparison operators such as $|$ (divides) are possible when dealing with integers.

### E.2 Tactics

Equations allows to deduce equalities and inequalities from simpler subgoals, using elementary rules (i.e. tactics). The environment contains two types of rules: transformations, which consist in matching a pattern in an expression and replacing it by an equivalent expression; and assertions, which consist in asserting that an expression is true. Both types of rules can have assumptions.

**Transformation rules**   A transformation rule (`TRule`) consists in a set of two expressions, $L$ and $R$, equivalent under a set of assumptions $S$. For instance `TRule`$(A + B, B + A)$ is the transformation rule stating the commutativity of the addition, namely that $A + B = B + A$ for any expressions $A$ and $B$. Note that in this case, the set of assumption $S$ is empty as the equality always holds. Another example is `TRule`$(\sqrt{A^2}, A, [A \geq 0])$ that states that $\sqrt{A^2} = A$ provided that $A \geq 0$.

Applying such a rule to an existing equation works as follows:

- matching a term $T$ in the expression that has the pattern of $L$
- identifying the matching variables and substituting them in $R$
- replacing $T$ by R in the input equation
- return the resulting equation with the set of hypotheses required for the transformation

For instance, if the input goal is:
$$\sqrt{(e^x)^2} = e^x$$

Applying `TRule`$(\sqrt{A^2}, A, [A \geq 0])$ on this expression will result in two subgoals:

- The same expression, where $\sqrt{A^2}$ has been replaced by $A$: $e^x = e^x$
- The hypothesis required for the assumption to hold: $e^x \geq 0$

More generally, a transformation rule will result in $N + 1$ subgoals, where $N$ is the number of hypotheses required by the rule.

**Assertion rules**   An assertion rule (`ARule`) expresses the fact that an expression is true, provided some hypotheses. It is represented by a main expression, and a set of assumptions sufficient for the main expression to hold. For instance, the rule `ARule`$(A \leq C, [A \leq B, B \leq C])$ states the transitivity of the partial order $\leq$, i.e. $A \leq C$ provided that there exists an expression $B$ such that $A \leq B$ and $B \leq C$.

Assertion rules do not always have hypotheses, for instance the reflexivity rule $\texttt{ARule}(A = A)$, or the rule $\texttt{ARule}(e^A > 0)$ stating that $e^A$ is positive, for any real value $A$. Note that the two subgoals generated in the previous paragraph ($e^x = e^x$ and $e^x > 0$) can be respectively solved by these two assertion rules (i.e. by matching $A = e^x$ and $A = x$).

Unlike transformation rules that always result in at least one subgoal (the initial expression on which we applied the transformation), assertion rules will only generate $N$ subgoals, where $N$ is the number of hypotheses. As a result, being able to apply an assertion rule without hypotheses to an expression is enough to close (e.g. solve) the goal. Assertion rules are in fact very similar to rules in Metamath.

In Table 6, we provide the number of Equations rules in different categories. Some examples of transformation and assertion rules are given in Table 7.

Table 6: **Number of Equations rules in each category.**

| Rule type | Basic | Exponential | Trigonometry | Hyperbolic | All |
|---|---|---|---|---|---|
| Transformation | 74 | 18 | 9 | 8 | 109 |
| Assertion | 90 | 11 | 9 | 0 | 110 |
| Total | 171 | 29 | 18 | 11 | 219 |

Table 7: **Trigonometric rules** accessible by the model. The model only has access to these elementary rules when proving statements from *Identities*. In particular, it cannot use more involved theorems such as $\cos^2(x) + \sin^2(x) = 1$.

| Transformation rules | Assertion rules |
|---|---|
| $\sin(0) = 0$ | $|\cos(A)| \leq 1$ |
| $\cos(0) = 1$ | $|\sin(A)| \leq 1$ |
| $\sin(\frac{\pi}{2}) = 1$ | $|\sin(A)| \leq |A|$ |
| $\cos(\frac{\pi}{2}) = 0$ | $A = B \implies \sin(A) = \sin(B)$ |
| $\sin(-A) = -\sin(A)$ | $A = B \implies \cos(A) = \cos(B)$ |
| $\cos(-A) = \cos(A)$ | $\sin(A) \neq \sin(B) \implies A \neq B$ |
| $\cos(A) \neq 0 \implies \tan(A) = \frac{\sin(A)}{\cos(A)}$ | $\cos(A) \neq \cos(B) \implies A \neq B$ |
| $\sin(A + B) = \sin(B)\cos(A) + \sin(A)\cos(B)$ | $A = B, \cos(A) \neq 0 \implies \tan(A) = \tan(B)$ |
| $\cos(A + B) = \cos(A)\cos(B) - \sin(A)\sin(B)$ | $\tan(A) \neq \tan(B), \cos(A)\cos(B) \neq 0 \implies A \neq B$ |

### E.3 Proving a statement with Equations

In order to prove a theorem with Equations, the user (or automated prover) has to apply tactics on the current expression. A tactic can correspond either to a transformation rule, or to an assertion rule.

For transformation rules, the model needs to provide:

- the rule (using a token identifier)
- the direction in which the rule is applied (a Boolean symbol, for forward or backward)
- an integer that represents the position where the rule is applied
- an optional list of variables to specify (c.f. paragraph below)

The direction of the rule indicates whether we want to transform $L$ by $R$ or $R$ by $L$ (e.g. replace $A$ by $\sqrt{A^2}$, or the opposite). The position where the rule is applied is given by the prefix decomposition of the input expression. For instance, the prefix notation of $(x + y) + 1$ is given by $\texttt{+ + x y 1}$. Applying the commutativity rule $A + B = B + A$ to the expression in position 0 will result in $1 + (x + y)$. Applying it in position 1 will result in $(y + x) + 1$, since the rule was applied to $(x + y)$. Note that for the commutativity rule, the direction in which we apply the rule does not matter. The list of variables to specify is required when variables in the target patterns are absent from the source pattern. For instance, applying the transformation rule $\texttt{TRule(A,A+B-B)}$ in the forward direction will require to provide the value of $B$.

For assertion rules, the format is simpler. We no longer need to specify a direction or a position (the position is always 0 as the assertion statement must match the expression to prove). We just need to provide:

- the rule (using a token identifier)
- an optional list of variables to specify

In this case, the list of variables to specify corresponds to variables that appear in hypotheses and cannot be inferred from the main expression. For instance, to apply the assertion rule $A \leq B, B \leq C \implies A \leq C$, we need to specify the value of $B$. We will then be left with two subgoals: $A \leq B$ and $B \leq C$.

Proving a statement in Equations requires to recursively apply tactics to unproved subgoals, until we are left with no subgoals to prove.

An example of proof-tree in Equations is shown in Figure 6. Figure 8 shows an example proof of the statement $(\mathbf{x} - \mathbf{y}) - (\mathbf{x} + \mathbf{y}) + \mathbf{2y} = \mathbf{0}$ using rules from the environment. Although simple, this statement requires 22 proof steps and highlights the depth required to prove complex mathematical identities when using elementary proof steps.

| Statement to prove | Rule used |
|---:|---|
| $(x - y) - (x + y) + 2y = 0$ | $A - B = A + (-B)$ |
| $(x - y) + (-(x + y)) + 2y = 0$ | $-(A + B) = (-A) + (-B)$ |
| $(x - y) + ((-x) + (-y)) + 2y = 0$ | $A + (B + C) = A + B + C$ |
| $(x - y) + (-x) + (-y) + 2y = 0$ | $A + (-B) = A - B$ |
| $(x - y) + (-x) - y + 2y = 0$ | $A + (-B) = A - B$ |
| $(x - y) - x - y + 2y = 0$ | $\text{int}(a + b) = \text{int}(a) + \text{int}(b)$ |
| $(x - y) - x - y + (1 + 1) \times y = 0$ | $A \times B = B \times A$ |
| $(x - y) - x - y + y \times (1 + 1) = 0$ | $A \times (B + C) = A \times B + A \times C$ |
| $(x - y) - x - y + y \times 1 + y \times 1 = 0$ | $A \times 1 = A$ |
| $(x - y) - x - y + y + y \times 1 = 0$ | $A - B = A + (-B)$ |
| $(x - y) - x + (-y) + y + y \times 1 = 0$ | $A + B = B + A$ |
| $(x - y) - x + y + (-y) + y \times 1 = 0$ | $A + (-B) = A - B$ |
| $(x - y) - x + y - y + y \times 1 = 0$ | $A - A = 0$ |
| $(x - y) - x + 0 + y \times 1 = 0$ | $A + 0 = A$ |
| $(x - y) - x + y \times 1 = 0$ | $A - B = A + (-B)$ |
| $x + (-y) - x + y \times 1 = 0$ | $A + B = B + A$ |
| $(-y) + x - x + y \times 1 = 0$ | $A - A = 0$ |
| $(-y) + 0 + y \times 1 = 0$ | $A + 0 = A$ |
| $(-y) + y \times 1 = 0$ | $A + B = B + A$ |
| $y \times 1 + (-y) = 0$ | $A + (-B) = A - B$ |
| $y \times 1 - y = 0$ | $A \times 1 = A$ |
| $y - y = 0$ | $A - A = 0$ |
| $0 = 0$ | |

Figure 8: **Proof of the identity $(\mathbf{x} - \mathbf{y}) - (\mathbf{x} + \mathbf{y}) + \mathbf{2y} = \mathbf{0}$ with elementary rules.** In this example we provide at each step the current goal and the rule that is used to obtain the next goal. This example shows how difficult it can be to prove even simple statements in Equations as they may require a significant number of proof steps (22 in that case). This explains that proving more involved statements from *Identities* such as $\cosh(3x) = 4\cosh(x)^3 - 3\cosh(x)$ can require to generate very large proof trees.

## E.4 True expressions and numerical evaluation

Some theorems are trivial, either because their statements match the pattern of an assertion rule that has no assumptions (e.g. $x^2 \geq 0$ or $e^{y-x} \neq 0$), or because they do not contain any variable and an exact numerical evaluation can attest that they are true (e.g $(-1)/2 < 6$ or $1 - 7/4 = -6/8$).

To prevent the model from wasting budget in "uninteresting" branches, we automatically discard generated subgoals that can be trivially verified. However, we only perform numerical verification of expressions without variables when they exclusively involve rational numbers. For instance, we will automatically close subgoals such as $5 < (-3)^2$ or $\frac{1}{2} > \frac{1}{4}$, but not $e^1 < e^2$ or $\cos(3) \neq 0$. To prove that $e^1 < e^2$ the model will need to use, for instance, an assertion rule such as $A < B \implies e^A < e^B$ ($1 < 2$ will then be closed automatically).

In early implementations of the Equations environment, we found that the model was able to leverage vulnerabilities in the environment to reach a 100% accuracy and to prove any statement. These issues where coming from numerical approximations that were initially allowed during the numerical verification of constant expressions (c.f. Section E.4). To prevent these vulnerabilities, we restricted the numerical verification to rational expressions, in order to have an exact numerical evaluation and to avoid errors due to approximations. We give two examples of vulnerabilities found by the model when expressions were verified with an approximate numerical evaluation.

In Figure 9, the model manages to prove that $2 = 3$ by using the injectivity of the exponential function, and the fact that for NumPy, $\exp(-\exp(\exp(2))) = \exp(-\exp(\exp(3)))$. Evaluating the left and the right-hand side both numerically evaluate to $0.0$, and the environment incorrectly considered the expression to be valid.

In Figure 10, the model manages to prove that $0 \neq 0$ by first proving that $\cos(\pi/2) \neq 0$, and combining this result with the fact that $\cos(\pi/2) = 0$. The imprecision came from the NumPy approximation of $\cos(\pi/2)$ in $6.123 \times 10^{-17}$, and in particular the fact that $(((\cos(\pi/2)^{0.5})^{0.5})^{0.5}) \approx 9.4 \times 10^{-3}$, which was considered large enough by our threshold to be considered non-zero. By using this approximation, and the assertion rule $\sqrt{A} \neq 0 \implies A \neq 0$, the model was able to conclude that $(((\cos(\pi/2)^{0.5})^{0.5})^{0.5}) \neq 0 \implies \cos(\pi/2) \neq 0 \implies 0 \neq 0$.

$$
\begin{aligned}
& 2 = 3 && \text{Statement to prove} \\
\iff\; & e^2 = e^3 && \text{Rule: } A = B \iff e^A = e^B, \\
\iff\; & e^{e^2} = e^{e^3} && \text{Rule: } A = B \iff e^A = e^B, \\
\iff\; & -e^{e^2} = -e^{e^3} && \text{Rule: } A = B \iff -A = -B, \\
\iff\; & e^{-e^{e^2}} = e^{-e^{e^3}} && \text{Rule: } A = B \iff e^A = e^B, \\
\iff\; & 0 = 0 && \text{Numerical evaluation}
\end{aligned}
$$

Figure 9: **False "proof" of $2 = 3$ found by the model when allowing numerical approximation to verify constant expressions.** The model noticed that $\exp(-e^{e^2}) = \exp(-e^{e^3})$ is considered true by NumPy (as the left and the right hand side are both approximated to 0.0) to conclude that $2 = 3$ using the injectivity of the exponential function.

## E.5 Random theorem generator

While Metamath and Lean come with a collection of annotated theorems that can be used for training, Equations does not have an equivalent of manually proved statements. Instead, we generate a supervised training set of theorems to pretrain the model before we start the online training. We propose two simple generation procedures: a random walk, and a graph generation approach.

**Random walk generation** The random walk is the simplest way to generate a theorem. We start from an initial expression $A_0$ and a set of initial hypotheses, both randomly generated following the method of Lample and Charton [40]. Then, we randomly apply an admissible transformation rule on

$$0 \neq 0 \iff \cos \frac{\pi}{2} \neq 0 \iff \left( \cos \frac{\pi}{2} \right)^{0.5} \neq 0$$

$$\iff \left( \left( \cos \frac{\pi}{2} \right)^{0.5} \right)^{0.5} \neq 0 \iff \quad \ldots \quad \iff \left( \left( \left( \cos \frac{\pi}{2} \right)^{0.5} \right)^{\cdots} \right)^{0.5} \neq 0$$

Figure 10: **False "proof" that $0 \neq 0$ found by the model when allowing numerical approximation to verify constant expressions.** Since $\cos(\frac{\pi}{2})$ evaluates to $6.123 \times 10^{-17}$ in NumPy (and not exactly to 0), the model found that for any tolerance threshold applying the assertion rule $\sqrt{A} \neq 0 \implies A \neq 0$ enough times lead to an expression where the left-hand side evaluates numerically to a strictly positive value. In particular, $(((\cos(\frac{\pi}{2})^{2^{-3}}) \approx 9.4 \times 10^{-3}$, which was considered large enough by our threshold to be considered non-zero. After that, any expressions $A$ and $B$ can be shown to be equal using the assertion rule $(A \times C = B \times C \ \wedge \ C \neq 0) \implies A = B$ where $C$ is chosen to be 0 since $0 \neq 0$.

$A_0$ to get an equivalent expression $A_1$. The process is repeated, to get a sequence $A_0, A_1, \ldots, A_N$ of equivalent expressions. The final theorem consists in proving that $A_0 = A_N$, and the proof corresponds to the sequence of rules sequentially applied. To increase the diversity of generations, and to avoid sampling only rules without or with simple assumptions, we add a bias in the random sampling of rules to over-sample the underrepresented ones.

**Graph generation** Because of the simplicity of the random walk approach, the generated theorems tend to be easy to prove, and the model quickly reaches a perfect accuracy on the generated theorems. Moreover, proofs generated by the random walk are only composed of transformation rules. To generate a more diverse set of theorems, we also use a graph generation procedure, that creates a large acyclic graph of theorems, where each node is connected to its children by a rule in the environment. To create such a graph, we proceed as follows. We first generate a set of initial hypotheses, and initialize the graph with a node for each hypothesis. We then randomly apply a transformation or assertion rule on nodes already in the graph.

For instance, if $A \leq B$ and $B \leq C$ are two nodes in the graph, then we can add the node $A \leq C$ using the assertion rule $A \leq B \wedge B \leq C \implies A \leq C$. If $x = y \times (z - 1)$ is a node in the graph, we can use the transformation rule $B \neq 0 \implies A/B = C \iff A = B \times C$ to add the node $x/y = z - 1$, provided that the node $y \neq 0$ is also in the graph. Required hypotheses that are trivially verifiable (e.g. $2 > 0$ or $e^{-x} > 0$) are automatically added to the graph.

### E.6 Translating Equations theorems to Lean

**Exporting theorems to Lean.** To enrich the existing Lean supervised dataset with synthetic data, we built a translator from Equations to Lean. Although Equations statements are easy to translate, proofs can only be translated if they involve rules that also exist in Lean. Since Equations is a modular environment where rules can be specified by the user, we created a collection of Equations rules from existing Mathlib statements. Synthetic theorems can then be generated using the random walk or random graph approaches described in Section E.5, and converted into Lean to augment the existing supervised dataset. Examples of randomly generated Lean proofs are provided in Figure 11.

```
1  theorem SYNTHETIC_0
2    (x1 x3 x4 : ℝ) :
3    ((0:ℝ) ≤ ((real.cos (real.cos ((-6:ℝ) / ((x1 - x4) / x3)))) / (2:ℝ))) :=
4  begin
5    apply norm_num.nonneg_pos,
6    apply half_pos,
7    apply real.cos_pos_of_le_one,
8    apply real.abs_cos_le_one,
9  end
10
11 theorem SYNTHETIC_1
12   (x1 x4 : ℝ)
13   (h₀ : ((x4 * (real.exp x1)) < 10)) :
14   ((-((abs ((x4 * (real.exp x1)) - 10)) / 2)) < ((abs (10 - (x4 * (real.exp x1)))) / 2)) :=
15 begin
16   have h₁ : ((0:ℝ) < ((abs ((x4 * (real.exp x1)) - 10)) / 2)),
17   apply half_pos,
18   apply abs_pos_of_neg,
19   apply sub_neg_of_lt h₀,
20   apply norm_num.lt_neg_pos _ _ h₁,
21   rw ← abs_sub_comm,
22   apply half_pos,
23   apply abs_pos_of_neg,
24   apply sub_neg_of_lt h₀,
25 end
```

Figure 11: **Example of a randomly generated theorems in Lean**. The theorems were initially generated in the Equations environment using rules from the Mathlib library, and converted to Lean.

**Importing rules from Mathlib.** To allow interfacing Equations and Lean, we automatically parsed Mathlib statements from the Lean library, and extracted theorems with a statement compatible with the Equations environment. Compatible theorems are converted into Equations transformation or assertion rules. Overall, we converted 1702 theorems from the Lean Library into our Equations environment. Details about the number of converted theorems are provided in Table 8.

Table 8: **Number of Equations rules converted from Lean**. The converted Lean theorems can be used to generate synthetic theorems within the Equations environment. The generated theorems can then in turn be converted back to Lean, along with their proofs. Some theorems are generic and can be applied to different types of variables (e.g. add_comm), and will appear in different categories. Overall, we automatically converted 1702 different Lean rules in our Equations environment.

| Rule type | Natural numbers | Integers | Real numbers |
|---|---|---|---|
| Transformation | 304 | 452 | 799 |
| Assertion | 314 | 292 | 407 |
| **Total** | 618 | 744 | 1206 |

### E.7  Examples of identities solved by the model on Equations

In Table 9, we give some examples of identities solved by the model. For each statement, we indicate the proof size and the proof depth, for the first proof found by the model, and for the optimal proof. We observe that the first proofs are sometimes very large, with more than 100 nodes, and that the model later manages to find shorter proofs as it improves.

Table 9: **Examples of identities solved.** Some of the 144 identities found by our model, in the order they were first solved. For each identity, we provide the size and the depth, both the for first proof, and for the minimal proof (i.e. the proof with the smaller number of steps) found during online training. The model found proofs with over 350 steps, some exceeding a depth of 100. After additional proof search, the model is often able to find shorter proofs. The proof of $\sin(2\pi + x) = \sin(x)$ requires a large number of steps, as the model can only use simple rules (e.g. the trigonometric rules provided in Table 7), and it does not have access to the value of $\sin(2\pi)$ or $\sin(\pi)$.

| Identity | Proof size | | Proof depth | |
| --- | --- | --- | --- | --- |
| | First | Best | First | Best |
| $\exp(-x)\exp(x-y) = \exp(-y)$ | 6 | 6 | 6 | 6 |
| $\cosh(-x) = \cosh(x)$ | 4 | 4 | 4 | 4 |
| $\sin(\pi/2 + x) = \cos(x)$ | 8 | 8 | 8 | 7 |
| $0 < x \implies 2\ln(\sqrt{x}) = \ln(x)$ | 16 | 3 | 7 | 3 |
| $\cos(\pi/2 - x) = \sin(x)$ | 19 | 11 | 19 | 10 |
| $\sin(\pi/2 - x) = \cos(x)$ | 14 | 10 | 14 | 10 |
| $\cos(x)^2 + \sin(x)^2 = 1$ | 13 | 11 | 13 | 10 |
| $\cos(x) = \cos(x/2)^2 - \sin(x/2)^2$ | 16 | 11 | 16 | 7 |
| $\sin(x+y) - \sin(x-y) = 2\sin(y)\cos(x)$ | 24 | 14 | 23 | 14 |
| $0 < x \implies 2x\cosh(\ln(x)) = x^2 + 1$ | 20 | 14 | 18 | 12 |
| $\tanh(x) = (\exp(x) - \exp(-x))/(\exp(x) + \exp(-x))$ | 46 | 23 | 30 | 11 |
| $\cos(x-y) + \cos(x+y) = 2\cos(x)\cos(y)$ | 33 | 19 | 33 | 13 |
| $\cosh(x) - \sinh(x) = \exp(-x)$ | 27 | 20 | 27 | 19 |
| $\cosh(x) - \sinh(x) = \frac{1}{\sinh(x)+\cosh(x)}$ | 55 | 38 | 40 | 20 |
| $\sin(2x) = 2\sin(x)\cos(x)$ | 27 | 15 | 19 | 8 |
| $\cos(2x) = 1 - 2\sin(x)^2$ | 130 | 27 | 118 | 21 |
| $\cosh(x-y) + \cosh(x+y) = 2\cosh(x)\cosh(y)$ | 84 | 31 | 84 | 29 |
| $\tanh(x) = (\exp(2x) - 1)/(\exp(2x) + 1)$ | 205 | 65 | 176 | 39 |
| $\sin(x) = 2\sin(x/2)\cos(x/2)$ | 29 | 17 | 21 | 8 |
| $\cos(2x) = 2\cos(x)^2 - 1$ | 72 | 26 | 68 | 19 |
| $\cos(x)^2 = 1 + \cos(2x)/2$ | 71 | 30 | 61 | 16 |
| $\sinh(x) = 2\sinh(x/2)\cosh(x/2)$ | 64 | 37 | 51 | 25 |
| $\sinh(2x) = 2\sinh(x)\cosh(x)$ | 71 | 34 | 61 | 24 |
| $\sinh(x+y) = \sinh(x)\cosh(y) + \cosh(x)\sinh(y)$ | 130 | 77 | 121 | 63 |
| $\cosh(x-y) = \cosh(x)\cosh(y) - \sinh(x)\sinh(y)$ | 90 | 66 | 75 | 56 |
| $\cos(x+y)\cos(x-y) = \cos(x)^2 - \sin(y)^2$ | 117 | 64 | 117 | 64 |
| $\sin(x+y)\sin(y-x) = \cos(x)^2 - \cos(y)^2$ | 118 | 64 | 118 | 63 |
| $|\sinh(x/2)| = \sqrt{(\cosh(x) - 1)/2}$ | 86 | 53 | 61 | 36 |
| $\sin(x+y)\sin(x-y) = \sin(x)^2 - \sin(y)^2$ | 183 | 66 | 183 | 65 |
| $\cosh(x)^2 = 1 + \cosh(2x)/2$ | 87 | 40 | 71 | 32 |
| $\cosh(2x) = 2\cosh(x)^2 - 1$ | 78 | 42 | 62 | 33 |
| $\cosh(2x) = \cosh(x)^2 + \sinh(x)^2$ | 97 | 72 | 80 | 64 |
| $\tanh(x) - \tanh(y) = \sinh(x-y)/(\cosh(x)\cosh(y))$ | 154 | 135 | 85 | 81 |
| $\tanh(x) + \tanh(y) = \sinh(x+y)/(\cosh(x)\cosh(y))$ | 162 | 144 | 95 | 91 |
| $\sqrt{1 + \sinh(x)^2} = \cosh(x)$ | 82 | 70 | 76 | 62 |
| $\sin(x)^3 = (3\sin(x) - \sin(3x))/4$ | 72 | 58 | 63 | 49 |
| $\sin(3x) = 3\sin(x) - 4\sin(x)^3$ | 80 | 56 | 71 | 47 |
| $\cosh(3x) = 4\cosh(x)^3 - 3\cosh(x)$ | 204 | 105 | 176 | 79 |
| $\cosh(x)^3 = (3\cosh(x) + \cosh(3x))/4$ | 162 | 106 | 137 | 79 |
| $\sin(4x) = \cos(x)(4\sin(x) - 8\sin(x)^3)$ | 73 | 73 | 60 | 60 |
| $\cos(\pi + x) = -\cos(x)$ | 148 | 28 | 118 | 9 |
| $\sin(\pi - x) = \sin(x)$ | 73 | 28 | 45 | 11 |
| $\cos(\pi/3) = \sin(\pi/6)$ | 26 | 17 | 26 | 17 |
| $\cos(\pi/4) = \sin(\pi/4)$ | 24 | 17 | 24 | 17 |
| $\cos(\pi/6) = \sin(\pi/3)$ | 22 | 17 | 22 | 17 |
| $\cos(2\pi + x) = \cos(x)$ | 125 | 70 | 37 | 18 |
| $\sin(2\pi + x) = \sin(x)$ | 353 | 69 | 62 | 16 |

## F    Example Lean proofs

In this section, we show examples of proofs found by our model.

```
1   theorem imo_1964_p1_2 (n : ℕ) : ¬7|2^n + 1 :=
2   begin
3     rw nat.dvd_iff_mod_eq_zero,
4     rewrite [nat.add_mod, nat.mod_eq_of_lt],
5     obviously,
6     apply nat.strong_induction_on n,
7     induction n,
8     {
9       intros n IH,
10      cases n,
11      norm_num,
12      cases n,
13      norm_num,
14      rw [nat.succ_eq_add_one, pow_succ],
15      rw [nat.succ_eq_add_one, pow_succ],
16      induction n,
17      norm_num,
18      rw [nat.succ_eq_add_one, pow_succ],
19      norm_num [nat.mul_mod, ←mul_assoc],
20      contrapose! IH,
21      refine ⟨n_n, nat.lt_succ_iff.mpr _, IH⟩,
22      exact nat.le_succ_of_le (nat.le_succ _),
23    },
24    exact n_ih,
25  end
```

Figure 12: **A proof of the `imo_1964_p1_2` problem found by our model.** The model shows that for any value of $n \in \mathbb{N}$, $2^n + 1$ is not divisible by 7, by showing that $2^n \bmod 7 + 1 \neq 0$, and $2^n \bmod 7 + 1 < 7$. The second part of the proof uses strong induction and the fact that $2^n \equiv 2^{n+3} \mod 7$. We provide a version of the proof that was automatically cleaned by removing unnecessary tactics and tactic arguments.

```
1   theorem imo_2001_p6
2     (a b c d : ℕ)
3     (h₀ : 0 < a ∧ 0 < b ∧ 0 < c ∧ 0 < d)
4     (h₁ : d < c)
5     (h₂ : c < b)
6     (h₃ : b < a)
7     (h₄ : a * c + b * d = (b + d + a - c) * (b + d - a + c)) :
8     ¬ nat.prime (a * b + c * d) :=
9   begin
10    contrapose h₄,
11    rw mul_comm,
12    simp [nat.prime, not_le_of_gt h₀.1, not_forall, not_le_of_gt h₃,
13          nat.mul_sub_right_distrib, nat.add_comm],
14    contrapose! h₄,
15    contrapose! h₄,
16    apply has_lt.lt.ne,
17    apply nat.lt_sub_right_of_add_lt,
18    nlinarith,
19  end
```

Figure 13: **A proof found by our model of another IMO problem in miniF2F.** Although the proof is valid, the statement is erroneous. The hypothesis $h_4 : b + d - a + c$ actually represents $\max(b + d - a, 0) + c$. This is due to Lean's `nat` type behaviour where $(a : \mathbb{N}) - (b : \mathbb{N}) = (0 : \mathbb{N})$ if $b \geq a$. This makes the exercise easier than it should be, and the proof is no longer valid on the fixed statement.