# OpenReview forum: "HyperTree Proof Search for Neural Theorem Proving"
_NeurIPS.cc/2022/Conference — NeurIPS 2022 Accept_

### Official Review · Reviewer_kFm7 · 2022-07-02

**Rating:** 4
**Confidence:** 3
**Soundness:** 2 fair
**Presentation:** 2 fair
**Contribution:** 3 good

**Summary:**

This paper proposes HyperTree Proof Search(HTPS), an AlphaZero-like proof search algorithm that can automatically solve input theorems by generating a sequence of suitable tactics without human interaction.
This prover contains two parts, the first is the Alphazero-inspire architecture, the second is the transformer-based model. And based on the prover, also propose an online training architecture to train their model in a faster way.
This paper proposes a new environment named Equations which can easily prototype and understand the behavior of the models and proof search.
They claim to have 82.6% accuracy outperforming the previous state of the art of 56.5% by GPT-f in Metamath theorems dataset.
However, a major concern is its presentation, which is unclear and hard to follow and be convinced (e.g., many are undefined or not described well as described below). I would raise my grade, if the authors can address these comments well.


**Questions:**

Questions:
* The total number of tactics in some environments is unclear (Equations), which makes the rest of this paper hard to interpret. If the number of tactics is extremely large or unlimited, how is the policy trained? And how to sample it?
* Hypertree Proof Search is unclear. In line 150 figure 2. Didn’t explain how to update the v_T(g) when there exist more than one tactic. In line 663 “Algorithm 2”, t is undefined, and what is the v(g, t) (any different from v_T(g)) in the while loop.
* The Metamath theorems dataset which you got the state-of-the-art accuracy is unclear (where it is from or published).
* It is unclear how tactics are selected from the policy distribution.
* The authors should discuss more on the reasons for using encoder-decoder transformer instead of decoder-only transformer in Section D.2 Model architecture.
* Explain how different HTPS is from a traditional search, called proof number search (PNS) [Allis 1994]. And, you probably want to compare with a Alpha-Zero-based proof method using PNS [Wu et al  2022].
  * Allis, L. V., Searching for solutions in games and artificial intelligence, Ph.D. Thesis, University of Limburg, Maastricht, 1994.
  * L. V. Allis, M. van der Meulen, and H. J. van den Herik. Proof-number search. Artiﬁcial Intelligence, 66(1):91–124, 1994.
  * Ti-Rong Wu, Chung-Chin Shih, Ting Han Wei, Meng-Yu Tsai, Wei-Yuan Hsu, I-Chen Wu, "AlphaZero-based Proof Cost Network to Aid Game Solving", ICLR 2022, April 2022.

Other minor questions:
* Line 243, “Result can be found in Table 2” in the first column of Table 2, the Supervised model is not described or defined in this paper.
* Line 181, “During online training, we continue sampling from the supervised tasks which provide high-quality data” the authors should explain the supervised tasks contextes and the usage.
* Line 230, “also evaluate the pass@k by running k proof searches on the evaluated statements with the policy and critic obtained by online training.”, what does it mean by running k proof searches? Does it mean simulation count using the terminology of MCTS?
* Line 588, “The search policy seletes t2, leading to the set of subgoals {g0,g1}”, but in figure 2 the root node is selected t1 to expansion.
* Line 738-745, “sampling temperature used during decoding”, “Length penalty”, “Depth penalty”, the authors should give an example of how to use these parameters.
* Line 751, “is trained to output a sequence of the following format: LABEL MANDATORY_SUBSTS <EOU> LABEL_STATEMENT PREDICTABLE_SUBSTS <EOS>”, the authors should provides some examples of the output for better understanding.
* Line 938, “Example Lean proofs” the authors should explain the case(s) in more detail. The current way is just like dumping something for readers to interpret themselves.


**Limitations:**

Yes

**Strengths And Weaknesses:**

Strengths:
* It is interesting to use Alphazero-like algorithm for Automated Theorem Proving.
* Based on their HTPS and online training, they seem to improve the state of the art on the Lean-based miniF2F-curriculum dataset and on the Metamath theorems dataset.
* This paper proposes a new environment named Equations which can easily prototype and understand the behavior of the models and proof search. (However, this is not mentioned in Abstract.)

Weaknesses:
The major problem is the presentation which is unclear and hard to follow. This makes this paper not convincing.
* First, many important algorithms (such as expanding and back-propagation) are described in appendix, not in main paper. It is obviously not self-contained, and makes it hard to read this paper and catch the point.
  * It is even worse that many terms are defined in appendix, while being used in the main paper. For example, W(g,t), N(g,t).
  * The term the minimal proof is used in the main paper, but only described as “i.e. the proof with the smaller number of steps” in the appendix.
  * The output of the model for metamath is described in appendix D.4, not in the main text.
* Many are undefined or not decribed. For example,
  * In Algorithm 1, VC(g,t) seems to represent a virtual visit count which is undefined.
  * In Algorithm 2, I don’t see where t is from. In Table 2, it is not described about how “Supervised” is obtained.
  * The proposed model (Transformer) is not described clearly, making it hard to interpret. Especially what is the model's (Transformer) in/output in each environment.
  * See more in Questions in more detail.

---

> ### Author Response · Authors · 2022-08-01
> **Response to reviewer kFm7 -- part 1/2**
>
> We thank the reviewer for their thorough review and helpful feedback. We fully agree with the reviewer that the description of the Selection, Expansion, and Back-propagation phases are critical for understanding and should not have been delegated to the appendix. We moved them back to the main section of the paper, and rewrote some parts accordingly. We described the list of changes we made to improve clarity in the common response to reviewers, and try to address the reviewer’s questions below. Please let us know if some answers are unclear.
>
> ---
>
> Q: "The total number of tactics in some environments is unclear (Equations), which makes the rest of this paper hard to interpret. If the number of tactics is extremely large or unlimited, how is the policy trained? And how to sample it?"
>
> A: For all environments, we split tactics into sequences of tokens. For Lean, there are around 50k tokens in the vocabulary, and tactics are sometimes composed of hundreds of tokens, so the number of tactics is basically unlimited. We updated the paper with concrete examples of tokenized goals with tactics.
>
> The decoder generates the tactics one token at a time in an autoregressive fashion, starting from a start symbol token until an end symbol token is reached, as in Sutskever et al., 2014. This language model is trained by minimizing the cross-entropy of the decoded output with a supervised target as in Sutskever et al. 2014.
>
> ---
>
> Q: "Hypertree Proof Search is unclear. In line 150 figure 2. Didn’t explain how to update the v_T(g) when there exist more than one tactic. In line 663 “Algorithm 2”, t is undefined, and what is the v(g, t) (any different from v_T(g)) in the while loop."
>
> A: We apologize for this typo which indeed made the algorithm confusing. The pseudo-code now only uses v_T(g) and the tactic 't' used in the updates of W(g,t) and N(g,t) is properly defined. As v_T(g) is associated with a specific partial proof-tree T, it is only ever associated with the single tactic selected at this node of the proof-tree.
>
> ---
>
> Q: "The Metamath theorems dataset which you got the state-of-the-art accuracy is unclear (where it is from or published)."
>
> A: GPT-f and Holophrasm both report results on randomly generated train/valid/test splits, but unfortunately none of them released their splits, so we had to create our own. In the Appendix, we release the exact version (GitHub hash) of Metamath we used in the paper. To facilitate reproducibility and comparison with other approaches, we would be happy to release our dataset splits if the paper is accepted to avoid anonymity issues.
>
> ---
>
> Q: "It is unclear how tactics are selected from the policy distribution."
>
> A: For online HTPS search, the selected tactic corresponds to the argmax of the PUCT policy, mixing language model score and current value estimates. This has been clarified in the main text (section 3.1, paragraph "Selection").
>
> ---
>
> Q: "The authors should discuss more on the reasons for using encoder-decoder transformer instead of decoder-only transformer in Section D.2 Model architecture."
>
> We used an encoder-decoder architecture (instead of decoder only) initially because we are in the context of a seq2seq problem (goal to tactic) and that encoder-decoder is a more natural architecture for seq2seq problems. But the main motivation was to accelerate the decoding speed when generating tactics. We followed the ideas behind the paper of Kasai et al. (Deep Encoder, Shallow Decoder: Reevaluating Non-autoregressive Machine Translation) that suggests that an encoder-decoder model with a small decoder can perform as well as with a large decoder, while decoding much faster. In our experiments, we found that multiplying by 2 the encoder size barely affects the generation speed, but dividing by 2 the decoder size almost makes generation twice faster. Besides, we also found in early experiments that the performance was very similar with small and large decoders, so we kept this architecture. Therefore, a large decoder-only model will necessarily be slow to decode tactics. We clarified this in the “Model Architecture” section to explain this motivation.
>
> ---
>
> Q: "Explain how different HTPS is from a traditional search, called proof number search (PNS) [Allis 1994]. And, you probably want to compare with a Alpha-Zero-based proof method using PNS"
>
> A: We've expanded the Appendix section comparing our search algorithm to pre-existing search algorithms to include a discussion on Proof  Number Search and other algorithms using estimated remaining proof size as a target. To summarize, the main difference is that given the arities at AND nodes in our set-up, we can afford to select all children. In terms of selection heuristic, we balance the tactic language model prior with an estimated value representing the probability of solving all required leaves. This estimated value will exponentially decrease with the proof-number, since the product of probability will include more terms.

---

> > ### Author Response · Authors · 2022-08-01
> > **Response to reviewer kFm7 -- part 2/2**
> >
> > Q: "Line 243, “Result can be found in Table 2” in the first column of Table 2, the Supervised model is not described or defined in this paper."
> >
> > A: By supervised model we refer to the model trained without online learning (and therefore only finetuned on supervised datasets), this has been added to the beginning of section 4.1 describing fine-tuning.
> >
> > ---
> >
> > Q: "Line 181, “During online training, we continue sampling from the supervised tasks which provide high-quality data” the authors should explain the supervised tasks contextes and the usage."
> >
> > A: These tasks are indeed defined in the next section, but we believe it is clearer to describe both the search algorithm and how it is used for online training in the same section, leaving the description of the supervised datasets to the fine-tuning section. We believe this separates our method contributions (proof-search and associated online data extraction and training) from other similar work, which use a similar pre-training followed by fine-tuning set-up. With the main text reworked, we hope that the content of these dataset is now made clearer.
> >
> > ---
> >
> > Q: "Line 230, “also evaluate the pass@k by running k proof searches on the evaluated statements with the policy and critic obtained by online training.”, what does it mean by running k proof searches? Does it mean simulation count using the terminology of MCTS?"
> >
> > A: Here, pass@k is similar to the metric used in the GPT-f papers and follow-ups. Specifically, one proof-search runs until a given expansion budget is exceeded (e.g. if there are more than 1000 nodes in the hyper-graph). Note that unlike MCTS where one simulation leads to exactly one expansion, in our HTPS algorithm, one simulation may lead to several expansions (one for each partial proof-tree leaf). Thus, for pass@k, we start k separate proof-search, until each separately exceeds its expansion budget or finds a proof.
> >
> > ---
> >
> > Q: "Line 588, “The search policy seletes t2, leading to the set of subgoals {g0,g1}”, but in figure 2 the root node is selected t1 to expansion."
> >
> > A: This was indeed a typo, which is fixed in the updated version of the paper. Thank you for spotting this out.
> >
> > ---
> >
> > Q: "Line 738-745, “sampling temperature used during decoding”, “Length penalty”, “Depth penalty”, the authors should give an example of how to use these parameters."
> >
> > A: A higher length penalty encourages the model to generate short tactics (i.e. with a small number of tokens). Setting a depth penalty discourages the model to explore nodes deep in the hyper-graph, and favor in-breadth exploration over in-depth. Overall, these hyper-parameters lead to different graph explorations, which we empirically found to be beneficial at test time, as different theorems can require proofs with a very different structure, and this is a simple way to increase the diversity of proof-searches.
> >
> > ---
> >
> > Q: "Line 751, “is trained to output a sequence of the following format: LABEL MANDATORY_SUBSTS <EOU> LABEL_STATEMENT PREDICTABLE_SUBSTS <EOS>”, the authors should provides some examples of the output for better understanding."
> >
> > A: "Concrete examples were indeed missing. We added a few examples of tokenized goals / tactics in Appendix A of the paper, along with the detailed description of these environments."
> >
> > ---
> >
> > Q: "Line 938, “Example Lean proofs” the authors should explain the case(s) in more detail. The current way is just like dumping something for readers to interpret themselves."
> >
> > A: We agree with the reviewer that the proof dump, in and of itself, is not very enlightening, we had included a few comments in the caption of the first proof to describe the high level idea of the proof found by the model. Unfortunately, following a Lean proof is always difficult without interactively working with a formal prover to see the progressive modifications applied to the tactic state. Similarly, including a few selected goal states within the proof wouldn't be sufficient to understand the proof. However, adding the raw proof allows readers to copy-paste it into the Lean interactive prover to follow it step by step.

---

> > > ### Comment · Reviewer_kFm7 · 2022-08-09
> > > **Reply to Authors**
> > >
> > > Although the authors clarified some questions, several points are still not addressed or clarified. For example,
> > > * The Metamath theorems dataset which you got the state-of-the-art accuracy is still not clarified and not convincing.
> > > * It is still not explained about I/O of the transformer and how to transfer these tactic to the tactics distribution even when the number of the tactics is large or unlimited.
> > > * For PNS and related articles, you simply mention it without in-depth analysis.
> > >
> > > In general, it is hard to support the acceptance of this paper in the current presentation either. And, my score remains unchanged.

---

> > > > ### Author Response · Authors · 2022-08-09
> > > > **response**
> > > >
> > > > > The Metamath theorems dataset which you got the state-of-the-art accuracy is still not clarified and not convincing.
> > > >
> > > > As explained in the paper (and in the above response) the dataset is created using the same approach as GPT-f, i.e. by creating a random train/valid/test split among the theorems in the Metamath library.
> > > >
> > > > > It is still not explained about I/O of the transformer and how to transfer these tactic to the tactics distribution even when the number of the tactics is large or unlimited.
> > > >
> > > > We do not understand what you mean by `how to transfer these tactic to the tactics distribution`. As explained in the above response and in the paper, the goals and tactics are both sequences of tokens. The tactics are decoded token-by-token in an auto-regressive fashion, using the approach described in `Sequence to Sequence Learning with Neural Networks` (Sutskever et al. 2014). This is a standard technique since 2014 which is used by almost all neural generative models.
> > > >
> > > > > you simply mention it without in-depth analysis
> > > >
> > > > We are happy to cite PNS, but we unfortunately cannot make an in-depth analysis about it because this is beyond the scope of this paper.

---

### Official Review · Reviewer_pbWQ · 2022-07-05

**Rating:** 4
**Confidence:** 3
**Soundness:** 2 fair
**Presentation:** 1 poor
**Contribution:** 2 fair

**Summary:**

The authors study the neural theorem proving problem that learns to facilitate the search of theorems' proofs by neural networks. They incorporate an additional online-training procedure inspired by Alphazero into the previous transformer-based automated theorem prover. Experimental results show that their model outperforms the state-of-the-art on some benchmarks with some metrics.

**Questions:**

* Why AlphaZero-inspired methods are beneficial for theorem proving?
* Will online learning be useful in practice, e.g., in the single-query setting, where only one unseen theorem is present to prove in addition to the training split?
* Can Evariste outperforms other online-learning methods with large generative models?

**Limitations:**

The paper discussed the insufficiency of current theorem-proving methods in the long term.

**Strengths And Weaknesses:**

The paper studies an important problem, neural theorem proving. The experimental results show that they can achieve state-of-the-art performances with a 10x training time speed-up.

However, there are major concerns about method justifications, experimental design and comparison, and paper writing.
* It is vague what the novelty and contribution of the methods are.
    - The justification for introducing *AlphaZero*-inspired methods is confusing for me. I cannot see *two players* in the theorem-proving environment. All actions or actors should have the same goal, finding the proof for the theorem. I am aware of the discussion in related work, saying "we need one move (max) that leads to subgoals that are all proven (min)", but it doesn't make sense to me.
    - Online/reinforcement learning, itself, could be beneficial. But, there have been works introducing it to theorem proving [1], as well as combining it with generative modeling [2].
* It is unclear if online learning is useful in practice and if their method can improve upon other methods in the online learning setting.
    - For example, it is unclear if online learning could be beneficial for proving one single theorem. In that case, the method cannot share information among all theorems in the test split so that they cannot be used as a new kind of curriculum.
    - In addition, models have more information and takes longer during inference in the online-learning setting compared with the supervised-learning setting used in sota. It is thus unsure if the improvements come from online learning itself or from their method.
    - I would suggest authors compare with other online methods and also report performances in the single-query inference setting.
* The paper is generally not well written and hard to understand. For example, it's confusing how their methods could solve or alleviate the difficulties of neural theorem proving as stated in the introduction. And the method section omits descriptions of the whole pipeline (including pertaining and fine-tuning of the large generative model), while the dataset/environment part, on the other hand, is very long with lots of parsing details.

Therefore, overall, I think this paper is not ready for publication yet.

[1] Kaliszyk, Cezary, et al. "Reinforcement learning of theorem proving." Advances in Neural Information Processing Systems 31 (2018).

[2] Wang, Mingzhe, and Jia Deng. "Learning to prove theorems by learning to generate theorems." Advances in Neural Information Processing Systems 33 (2020): 18146-18157.

---

> ### Author Response · Authors · 2022-08-01
> **Response to reviewer pbWQ**
>
> We thank the reviewer for their review and feedback.
>
> Q: "The justification for introducing AlphaZero-inspired methods is confusing for me. I cannot see two players in the theorem-proving environment."
>
> A: We mention alphazero as a successful and famous example of tree-search combined with online-learning of a policy and value model. We believe our set-ups share many commonalities and specific differences between our set-up and AlphaZero are made precise in the introduction. A more complete introduction to AND/OR trees in the context of theorem proving can be found in Holophrasm; we didn't want to introduce this formalism in our paper since we instead chose to work on hypergraphs.
>
> ---
>
> Q: "Online/reinforcement learning, itself, could be beneficial. But, there have been works introducing it to theorem proving [1], as well as combining it with generative modeling [2]."
>
> A: Reference [1] here focuses on first order logic and uses much simpler models, whereas we evaluate our model in the context of more complex theorem provers. Evaluating our methods on the environment from this reference represents a significant amount of work infeasible within the timeframe of this rebuttal. Reference [2] improves upon Holophrasm results which are very far below those of GPT-f on Metamath, to which we compare in this paper and which we outperform significantly. However, we added all of these relevant references to the related work, we thank the reviewer for pointing them out.
>
> ---
>
> Q: “For example, it is unclear if online learning could be beneficial for proving one single theorem. In that case, the method cannot share information among all theorems in the test split so that they cannot be used as a new kind of curriculum.”
>
> A: We thank the reviewer for this suggestion. We indeed plan to test the online learning method in the single theorem instance setting in the future. However, to facilitate comparison and comparability, we decided to focus on the setting adopted by the most recent papers in the field (e.g. GPT-f, Thor), and where the task is to maximize the performance on a held-out test set (e.g. miniF2F-test).
>
> ---
>
> Q: “In addition, models have more information and takes longer during inference in the online-learning setting compared with the supervised-learning setting used in sota. It is thus unsure if the improvements come from online learning itself or from their method.”
>
> A: Our results already include an apples-to-apples comparison with the state of the art results from GPT-f when running online training on statements from minif2f-curriculum while reporting results on minif2f-valid/test. This experiment shows either a 10x speed-up for similar performances on miniF2F-test, or a 4% increase in performance using the same amount of compute. On minif2f-curriculum in our online learning setup, our model obtains an accuracy of 42.5%, a relative improvement of almost 40% compared to GPT-f.
> Moreover, we provide thorough ablations on Metamath to identify the impact of each component. First, our supervised model (without online training) evaluated with HTPS obtains an accuracy of 65.4%, outperforming the best-first search of GPT-f in the same setting (42.6%). Then, Figure 3 (right) compares online learning with pure inference, with a similar compute budget, and shows an absolute improvement of 12% accuracy in favor of online learning.
>
> ---
>
> Q: “I would suggest authors compare with other online methods and also report performances in the single-query inference setting."
>
> A: As mentioned previously, we agree with the reviewer that this is an interesting setting. However, running such experiments would be too compute intensive to fit within the timeframe of this rebuttal. Besides, the number of experiments in the paper being already quite large, we are afraid that it would dilute the message in the paper and make it too complicated to understand.
>
> ---
>
> Q: "The paper is generally not well written and hard to understand. For example, it's confusing how their methods could solve or alleviate the difficulties of neural theorem proving as stated in the introduction. And the method section omits descriptions of the whole pipeline (including pertaining and fine-tuning of the large generative model)"
>
> A: We have a paragraph as well as a Figure (Figure 7) summarising the full training pipeline in Section D.1. This pipeline being quite similar to the one used in AlphaZero and many other papers from the RL literature, we considered that it was not as critical as the sections in the main part of the paper. Similarly, the techniques we use for pre-training and fine-tuning are relatively common, so we left them in Section D.2 of the appendix. We agree with the reviewer that the description of the formal environments was too long given the amount of information in the paper, so we moved the Section 3 to the appendix, and added back details about the three phases of HTPS to the main paper

---

> > ### Comment · Reviewer_pbWQ · 2022-08-08
> > **Reply to Authors**
> >
> > I would like to thank the authors for their reply.
> >
> > I am glad to see the authors have removed the unsupported claim in their new version, such as
> > * the original third challenge in introduction: "Third, in Chess or Go, playing a sub-optimal move does not necessarily lead to losing the game, thus exploring these branches can provide information. In theorem proving, it is frequent to generate tactics that result in subgoals that can no longer be proved and on which the model will waste significant search budget.",
> >
> > but I am still confused by the two-player justification, such as
> > * "Theorem proving can be thought of as computing game-theoretic value for positions in a min/max tree: to prove a goal, we need one move (max) that leads to subgoals that are all proven (min). Noticing heterogeneity in the arities of min or max nodes, we propose a search method that goes down simultaneously in all children of min nodes, such that every simulation could potentially result in a full proof-tree.".
> >
> > Overall, I find it difficult to support acceptance of this paper, because
> > * For experiments on Lean,
> >   - the performance improvements can come from the implicit test-dataset curriculum (which may not exist in practical environments) since online-learning methods can share information during testing,
> >   - and similarly for the speed-up (which may not exist in the common practical single-query inference setting), as validated by the authors: "As mentioned previously, we agree with the reviewer that this is an interesting setting. However, running such experiments would be too compute intensive to fit within the timeframe of this rebuttal. "
> > * The experimental results on Metamath themselves may not be convincing enough to support acceptance,
> > * while the methods are not that novel (replacing previous methods with better generative models?) and not well described/motivated (two-player?) either.
> >
> > Taking the importance and the difficulties of the neural theorem proving problem into account, I will keep my original score as "4: Borderline reject".

---

> > > ### Author Response · Authors · 2022-08-08
> > > **response**
> > >
> > > > The experimental results on Metamath themselves may not be convincing enough to support acceptance,
> > >
> > > Could the reviewer please elaborate on what kind of results would be convincing to support acceptance?
> > > On Metamath, our supervised model trained exclusively on human data obtains 61.2% accuracy on the Metamath test set, which is to compare with the 42.6% of GPT-f. This difference is solely due to our search algorithm, and not to the online training. In fact, our supervised model (trained with 400 GPU hours) even outperforms the 56.2% of GPT-f trained with expert iterations (and with 20 000 GPU hours). Moreover, our model trained with online training obtains 72.4% on the test set.
> > >
> > > > the performance improvements can come from the implicit test-dataset curriculum
> > >
> > > The numbers reported in the paper on the minif2f-curriculum dataset are reported in the exact same setting as GPT-f.
> > >
> > > > and similarly for the speed-up
> > >
> > > Similarly for the speed-up.
> > >
> > > > while the methods are not that novel (replacing previous methods with better generative models?)
> > >
> > > The previous works we compare against in the paper all use the exact same generative models as we do, i.e. a vanilla transformer model. In fact, our initial supervised models fine-tuned on human data are probably inferior to GPT-f (which inherits from the infrastructure and pre-training of GPT-3 which is much more advanced than ours), and the algorithms and methods presented in the paper is precisely what allows us to outperform the previous SOTA.

---

### Official Review · Reviewer_pUNj · 2022-07-08

**Rating:** 7
**Confidence:** 4
**Soundness:** 4 excellent
**Presentation:** 2 fair
**Contribution:** 4 excellent

**Summary:**

The paper presents a novel online, search-based method, similar to AlphaZero, to automatically prove mathematical theorems, in 3 different environments, including miniF2F. This new algorithm yields state-of-the-art results across the board, outperforming GPT-f by a significant margin. Further, the paper provides extensives ablations to motivate the final hyperparameter and algorithmic choices.

**Questions:**

1. For the training of the critic, a categorical loss with 2 labels is used. I wonder if simple regression, or a bucketed categorical loss have been tried, and whether they perform worse or better?
2. I don't think I've seen the sequence length the encoder and decoder are trained with, did I miss it or is it an oversight on the authors' part? Given the compute-intensive aspect of HTPS, and for reproducibility purposes, I think this information should be reported.
3. Reading the paper, my understanding is that the training is done on (goal, tactic) pairs, rather the the trajectories of a full proof. Are full proofs simply too long to fit a transformer model context? Running search on full trajectories is possible with transformers (caching keys and values), as is done in https://arxiv.org/abs/2104.05336.
4. One is almost surprised by the relatively small size of the models. While of course it's not possible for every lab to use much larger models, the general trend is that scale delivers significant gains. If it's not financially possible to use a larger model, it would at least be useful to know how smaller scale models perform, so we have an idea of the gains scale can deliver.


**Limitations:**

Limitations are adequately reported in the paper.

**Strengths And Weaknesses:**

This is a very nice piece of work, both original and significant, with potential for a large impact on the machine learning community and beyond. The performance gains over the previous state of the art are impressive, the methods are well-motivated, the experiments are well done and reported.

My main issue with it is that while it is well-written, it's actually fairly hard to follow. The explanation for that is that due to lack of space the authors have relegated most of the relevant information to the appendix. I think presenting the work in this fashion is very suboptimal and that a journal version would be a lot better for readers (and thus more impactful in the end).
In its current form, the paper is hard to follow and thus almost average, whereas in a journal form it could really be an excellent contribution.

Another relative weakness of the paper is that the relative work (even in its extended version in the Appendix) is too succinct.

Finally, a scaling study is missing. While using larger models would be expensive, results with smaller models would help a lot with understanding the gains that scale could yield.

---

> ### Author Response · Authors · 2022-08-01
> **Reviewer pUNj response**
>
> We thank the reviewer for their review and valuable feedback.
>
> We agree with the reviewer that the paper in its original version was hard to follow, and made some adequate changes in the updated version of the paper. Notably, we reworked the overall organisation to make the paper more self-contained and less dependent on the appendix.
>
> -------
>
> Q: For the training of the critic, a categorical loss with 2 labels is used. I wonder if simple regression, or a bucketed categorical loss have been tried, and whether they perform worse or better?
>
> A: The bucketed categorical loss is a good idea. In fact, we experimented with a bucketed proof length prediction similar to "Formal mathematics statement curriculum learning" by Polu et al, but it didn't provide improvements in our set-up, so we continued our work with the binary loss described in the paper for simplicity.
>
> -------
>
> Q: I don't think I've seen the sequence length the encoder and decoder are trained with, did I miss it or is it an oversight on the authors' part? Given the compute-intensive aspect of HTPS, and for reproducibility purposes, I think this information should be reported.
>
> A: For the encoder, we use a maximum length of 2048 tokens. For the decoder, we found that setting up smaller maximum tactic lengths significantly speeds up decoding. As a result, we  use a maximum decoding length of 512 for Metamath, 128 for Lean, and 32 for the Equation environment. We found that these maximum lengths allow to generate more than 99% of tactics in each environment. These decoding lengths have been added to the relevant appendix where each environment is described in details. We thank the reviewer for the suggestion.
>
> -------
>
> Q: Reading the paper, my understanding is that the training is done on (goal, tactic) pairs, rather the the trajectories of a full proof. Are full proofs simply too long to fit a transformer model context? Running search on full trajectories is possible with transformers (caching keys and values), as is done in https://arxiv.org/abs/2104.05336.
>
> A: The reviewer is correct, we only condition on the current goal. States can be quite large, so conditioning on the full set of explored nodes would be technically difficult. Concurrent work by Jiang et al. on Isabelle suggests that conditioning on the previous state (as is done in chess, for example) already leads to improved performances. This is definitely something that we will be doing in the future.
>
> -------
>
> Q: One is almost surprised by the relatively small size of the models. While of course it's not possible for every lab to use much larger models, the general trend is that scale delivers significant gains. If it's not financially possible to use a larger model, it would at least be useful to know how smaller scale models perform, so we have an idea of the gains scale can deliver.
>
> A: The scaling study is a great suggestion. We target similar model sizes to GPT-f, primarily to make comparison easier, and also because the authors of GPT-f reported that scaling up didn't lead to improved performances. However, it is possible that smaller models could trade-off accuracy for decoding speed, and result in a similar final performance. This is definitely something we will investigate, as we found that the tactic decoding speed could be a bottleneck in our Metamath and Equation environments, but we must unfortunately leave this to future work, since the rebuttal period doesn't allow enough time to run such experiments.

---

> > ### Comment · Reviewer_pUNj · 2022-08-08
> > **Thanks for the responses!**
> >
> > The rebuttal answers my questions in a satisfactory manner. I'm not upping my recommendation because my main issue was with the presentation, which I cannot control without having a proper look at the new version.
> > I maintain that this is a good paper, as good clean execution can be as important as novelty.

---

> > > ### Author Response · Authors · 2022-08-08
> > > **response**
> > >
> > > We thank the reviewer for their comments on our paper!
> > >
> > > The new version of the paper that we uploaded on August 2nd should contain all the modifications and the new presentation / organisation.
> > > We believe the reviewers have access to the last version of the paper on OpenReview and should be able to see the differences. Please let us know if this not the case.

---

### Official Review · Reviewer_UdaU · 2022-07-11

**Rating:** 5
**Confidence:** 4
**Soundness:** 3 good
**Presentation:** 3 good
**Contribution:** 2 fair

**Summary:**

The paper proposes a search algorithm called HyperTree Proof Search (HTPS) inspired by the recent success of AlphaZero. The search algorithm is combined with a transformer-based policy for choosing tactics to prove theorems in an interactive theorem prover. The resulting model is shown to significantly outperform the previous state of the art by GPT-f, which also leverages language models but only uses relatively naive search strategies (based on cumulative log probabilities of tactics). The experiments and ablation studies clearly validate the benefits of the proposed search algorithm.

**Questions:**

There are many missing citations along the lines of reinforcement learning for theorem proving, such as [Kaliszyk et al](https://arxiv.org/abs/1805.07563) and [Crouse et al](https://arxiv.org/abs/1911.02065). The reviewer suggests at least including [TacticZero](https://proceedings.neurips.cc/paper/2021/hash/4dea382d82666332fb564f2e711cbc71-Abstract.html) in the section of related work, as both have a focus on learning search strategies.

**Limitations:**

Yes,  the authors adequately addressed the limitations.

**Strengths And Weaknesses:**

Quality and clarity:

- The experiments are well-designed and the empirical results are very good. It is clear that this paper embodies a significant amount of work. There is also a great amount of computational resources consumed to achieve the good performance.
- The paper is well-written and easy to follow. It is a very good aggregation of existing techniques in ML for theorem proving that ultimately achieves the new state-of-the-art performance.

Originality:

- My major concern is the novelty. The concept of HyperTree Proof Search itself is not new. The framework of hyper tree proof search is very similar to what has been proposed in the [TacticZero](https://proceedings.neurips.cc/paper/2021/hash/4dea382d82666332fb564f2e711cbc71-Abstract.html) paper. The difference is that this paper learns a critic for the nodes with MCTS, while [TacticZero](https://proceedings.neurips.cc/paper/2021/hash/4dea382d82666332fb564f2e711cbc71-Abstract.html) learns a value function for the nodes end-to-end through online policy gradient.
- The usage of a transformer-based language model for policy has also been studied in a series of GPT-f papers, and recently in papers (e.g., [Thor](https://arxiv.org/abs/2205.10893)) targeting provers other than Lean or Metamath. To my knowledge, the combination of using language models for policy and learning a proof search strategy is indeed novel. However, I am not sure whether this level of novelty is significant enough for NeurIPS.

---

> ### Author Response · Authors · 2022-08-01
> **Reviewer UdaU response**
>
> We thank the reviewer for their review and helpful feedback.
>
> Q: "To my knowledge, the combination of using language models for policy and learning a proof search strategy is indeed novel. However, I am not sure whether this level of novelty is significant enough for NeurIPS."
>
> A:  From our perspective, the scientific novelty of our contribution lies in 3 different aspects. First, the combination of MCTS-type search with language models is indeed not documented in the literature. The online training procedure that is made possible by this setup not only leads to a significantly better overall performance, but also allows the offline model itself to obtain SOTA results in several benchmarks, as is illustrated in Tables 2 and 3. The fact that our approach works well on different proving environments seems to us worthy of scientific interest and will make future performance comparisons easier. Moreover, we provide detailed ablations of our method, to show the precise impact of each of its components in Section 6.2. Last but not least, we introduce Equations, a new framework for theorem proving. This framework, that we used as a testbed, provides significant insights compared to other environments and creates a simple toy environment for Theorem Proving. We are planning to open-source this environment which we hope will accelerate further research in this domain.
>
> -----------
>
> Q: "There are many missing citations along the lines of reinforcement learning for theorem proving, such as Kaliszyk et al and Crouse et al. The reviewer suggests at least including TacticZero in the section of related work, as both have a focus on learning search strategies. The usage of a transformer-based language model for policy has also been studied in a series of GPT-f papers, and recently in papers (e.g., Thor) targeting provers other than Lean or Metamath."
>
> A: We've added the suggested works to the related work section. Their omission was an oversight from our part, thank you for pointing them out. The Thor paper on the Isabelle prover was pre-published a week after we submitted our own paper to this conference.
> The search method presented in TacticZero is quite far from our own: TacticZero selects one goal at a time, using a softmax on the estimated value of goals within fringes. In comparison, each selection in our proposed HTPS search selects a subset of goals that, if closed, would close the root. Our selection also uses a critic model, but also relies on an optimistic upper-bound to balance exploration and exploitation. This is in addition to using a very different model with different training requirements, we use large transformer models on tokenized goal state / tactics, whereas TacticZero uses a RNN on featurized goals to predict a structured policy.

---

> > ### Comment · Reviewer_UdaU · 2022-08-07
> > **Response to the authors**
> >
> > Thanks for the response!
> > > The search method presented in TacticZero is quite far from our own: TacticZero selects one goal at a time, using a softmax on the estimated value of goals within fringes. In comparison, each selection in our proposed HTPS search selects a subset of goals that, if closed, would close the root.
> >
> > This isn’t quite accurate. Before selecting one goal, TacticZero first selects a fringe, which in the context of TacticZero is exactly a subset of goals that if closed, would close the root (see figure 1 of the paper). If in terms of your figure 2, each of the sets {$g_5$}, {$g_0,g_1$}, {$g_6, g_7$} and {$g_2, g_3, g_4$} would be a fringe. It is true that after selecting the set TacticZero only expands one goal within it while HTPS expands all of them, but this isn’t a huge difference. In fact, expanding all the goals might be inefficient, because the expansion of the first goal could result in something that’s already hopeless.
> >
> > Please feel free to clarify further if there is any misunderstanding.

---

> > > ### Author Response · Authors · 2022-08-08
> > > **response**
> > >
> > > Thank you for the response. We still believe the proposed search algorithm to be different from TacticZero. Whereas fringe selection in TacticZero is done by sampling from a softmax obtained by summing the open goal values within fringes, we select our "fringe" by following optimistic upper bounds in the hyper-tree. This is a critical difference that allows to balance exploration and exploitation, whereas, as far as we understand, TacticZero only follows its critic.
> > >
> > > Regarding the potential inefficiencies of expanding all goals, if a hopeless goal is produced at any other rank in the fringe than the first, our approach of expanding all goals prevents a fruitless exploration of a valid but useless first goal. In all of our attempts, we found expanding all goals in a fringe yielded better results than expanding them one by one.
> > >
> > > Moreover, as stated previously, our contribution doesn't only lie in the proposed algorithm, but also its extensive evaluation and ablations which significantly outperform the previous state of the art on multiple benchmarks.

---

> > > > ### Comment · Reviewer_UdaU · 2022-08-10
> > > > **Response**
> > > >
> > > > Thanks! I appreciate the clarification.
> > > > > This is a critical difference that allows to balance exploration and exploitation, whereas, as far as we understand, TacticZero only follows its critic.
> > > >
> > > > Following optimistic upper bounds isn’t the only way to balance exploration and exploitation. TacticZero follows its critic, but it does so stochastically, meaning that the Monte Carlo policy gradient algorithm it adopts naturally balances exploration and exploitation as well (and one can always further implement entropy regularization on top of that).
> > > >
> > > > I understand that the contribution doesn't only lie in the proposed algorithm, but I’m not yet fully convinced that the novelty of HTPS (as emphasized in the title) is that significant.

---

> > > > > ### Author Response · Authors · 2022-08-10
> > > > > **Response**
> > > > >
> > > > > We agree that there are similarities in the search algorithm, however, the differences that exist between them are critical. Typically, the best-first search algorithm by GPT-f is in fact closer to what we do than TacticZero, and yet in the same supervised setting (i.e. without online learning or iterative training) we obtain an accuracy of 65.4% Metamath, against 42.6% for GPT-f in a comparable setting, which is a huge difference.
> > > > >
> > > > > Besides, as mentioned by the reviewer, the novelty in our paper does not only lie in the search algorithm, but also in a new formal environment we propose, and in an overall methodology for online-training. There are important questions we answer in the paper, such as “which data should the provers send to the trainer during online training?”. We study this question thoroughly in our ablation experiments, and show that different strategies to filter the online training data can lead to very large differences of performance: 17% on Metamath, and more than 40% on Equations (c.f. Table 4).
> > > > >
> > > > > We also show that the setup we propose is quite efficient if we aim at maximizing the performance on a given subset of theorems from a new domain. Typically, in Equations, the online training strategy we propose can boost the accuracy by 42% (Figure 3, left) on a set of theorems with a new distribution.

---

### Author Response · Authors · 2022-08-01
**Common response to reviewers**

We thank all the reviewers for their valuable feedback and thorough comments.

Several reviewers pointed out that the paper was difficult to follow due to dependencies to the appendix. Thus, we reworked the overall organization to make the paper self-contained. Notably, we moved the description of the separate phases of HTPS into the main text. Some details about the formal environments that were not critical for the understanding of the paper were instead pushed to the appendix. Overall, we made a number of changes to improve clarity. We summarize these changes below:

- We moved detailed descriptions of each environment to the appendix and simplified the section on supervised dataset creation, making space for a more detailed description of the search algorithm in the main text.
- Addressed Reviewer kFm7 comments around missing notations and fixed errors and typos in the text and Algorithm 2.
- Clarified why we use an encoder-decoder architecture rather than decoder only.
- Clarified what we meant by “supervised model”, i.e. a model fine-tuned exclusively on human tactics, without online learning.
- Added some concrete examples of goals and tactics with their tokens, along with some information about their maximum length.
- Updated the related work section with several missing references pointed out by the reviewers. We've also expanded the "comparison with other search algorithm" Section in the appendix.

We replied individually to all reviewers to address their other questions and feedback.

---

### Meta-Review · Area_Chair_8NK2 · 2022-08-23

**Recommendation:** Accept
**Confidence:** Certain

**Metareview:**

This paper tackles automated theorem proving by combining MCTS with large language models. They show that with supervised pretraining, this combination can outperform a competitive alternative (GPT-f) using an order magnitude less compute, and that more dramatic improvements are possible if one allows online learning during testing. Reviewers said that the core ideas behind the work were not particularly new, but their combination was.

In summary, the paper is a new combination of old ideas that improves results on an important problem, and which has extensive experiments exploring the online test setting across 3 realistic domains of theorem proving. The online test setting applies when one has batches of difficult interrelated search problems, which is conceivably the case from many automated theorem proving settings. These factors suggest accepting the paper, but there were significant reservations from multiple reviewers. Fortunately these issues can be fixed: Although reviewers generally agreed that this was an improvement over the latest state of the art, at least quantitatively, and speculated that the architecture would likely be state of the art on other benchmarks as well, the following presentation issues should be addressed for the camera ready. There was significant confusion over whether _all_ of the paper's results were in the online setting, which could jeopardize the soundness of the quantitative results if baselines did not also use the online setting. Discussions clarified that this was not the case, but the manuscript should be revised to clearly and prominently state these facts to avoid misinterpretation. Referring to the work as AlphaZero-like promoted other confusions. Indeed, because the authors contrast online learning (what they do) with expert iteration (what AlphaZero does), and because this is not a two-player game, the authors are strongly encouraged to nix the AlphaZero analogy.

**Award:**

No

---

### Decision · Program_Chairs · 2022-09-14

Accept